# Seasonal variation of the sound-scattering zooplankton vertical distribution in the oxygen-deficient waters of the NE Black Sea

Alexander G. Ostrovskii[1], Elena G. Arashkevich[1], Vladimir A. Solovyev[1], Dmitry A. Shvoev[1]

[1] Shirshov Institute of Oceanology, Russian Academy of Sciences, 36, Nahimovskiy prospekt, Moscow, Russia, 117997

*Correspondence to*: Alexander G. Ostrovskii (osasha@ocean.ru)

**Abstract.** At the northeastern Black Sea research site, observations from 2010-2020 allowed us to study the dynamics and evolution of the vertical distribution of mesozooplankton in oxygen-deficient conditions via analysis of sound-scattering layers associated with dominant zooplankton aggregations. The data were obtained with profiler mooring and zooplankton net sampling. The profiler was equipped with an acoustic Doppler current meter, a conductivity-temperature-depth probe, and fast sensors for the concentration of dissolved oxygen [O2]. The acoustic instrument conducted ultrasound (2 MHz) backscatter measurements at 3 angles while being carried by the profiler through the oxic zone. For the lower part of the oxycline and the hypoxic zone, the normalized data of 3 acoustic beams (directional acoustic backscatter ratios, *R*) indicated sound-scattering mesozooplankton aggregations, which were defined by zooplankton taxonomic and quantitative characteristics based on stratified net sampling at the mooring site. The time series of ~14,000 *R*-profiles as a function of [O2] at depths where [O2] < 200 μM were analyzed to determine month-to-month variations of the sound-scattering layers. From spring to early autumn, there were two sound-scattering maxima corresponding to (1) daytime aggregations mainly formed by diel-vertical-migrating copepods *Calanus euxinus* and *Pseudocalanus elongatus* and chaetognaths *Parasagitta setosa,* usually at [O2] = 15-100 μM, and (2) persistent monospecific layer of the diapausing fifth copepodite stages of *C. euxinus* in the suboxic zone at 3 μM < [O2] < 10 μM. From late autumn to early winter, no persistent deep sound-scattering layer was observed. At the end of winter, the acoustic backscatter was basically uniform in the lower part of the oxycline and the hypoxic zone. The assessment of the seasonal variability of the sound-scattering mesozooplankton layers is important for understanding biogeochemical processes in oxygen-deficient waters.

## 1 Introduction

The main distinguishing feature of the Black Sea environment is its oxygen stratification with an oxygenated upper layer 80-200 m thick and the underlying waters containing hydrogen sulfide (Andrusov, 1890; see also review by Oguz et al., 2006). Early studies of the oxic zone indicated that the vertical distribution of zooplankton hinges on oxygen stratification (Nikitin, 1926; Petipa et al., 1960). Later, the dives of the manned research submersible Argus showed that the zooplankton vertical distribution was not uniform (Vinogradov et al. 1985; Flint 1989). In particular, the thin-layered structure of zooplankton distribution was observed by the Argus research pilot in the lower part of the oxic zone. Thereafter, zooplankton sampling

with a vertical resolution of 3-5 m using a 150-liter sampler with an attached conductivity-temperature-depth (CTD) probe
indicated that the daytime deep aggregations of the zooplankton populations were associated with layers of certain water
density (Vinogradov and Nalbandov, 1990; Vinogradov et al., 1992). The deeper zooplankton aggregation was formed by
the fifth copepodite stage of *Calanus ponticus* (old name of *C. euxinus*) and its lower boundary was at the specific density
surface $\sigma_\Theta = 15.9$, where the oxygen concentration was approximately 4 µM. The diapausing cohort of *C. ponticus* did not
perform vertical migrations and occupied the suboxic layer around the clock (Vinogradov et al., 1992). The accumulation of
a high lipid reserve, a decrease in the rate of oxygen consumption, and a delay in gonad development were defined as
characteristic features of diapausing *C. euxinus* (Vinogradov et al., 1992; Arashkevich et al., 1998; Svetlichny et al., 2002,
2006). The vertically migrating zooplankters (ctenophores *Pleurobrachia pileus*, chaetognaths *Parasagitta setosa*, and older
copepodites of *Pseudocalanus elongatus* and *C. euxinus*) formed daytime aggregations between isopycnals 15.7-15.5 and
15.4-14.9 and at an oxygen concentration of 11-40 µM. At night, the migrant zooplankters inhabited the upper layers and
peaked in the thermocline (Vinogradov et al., 1985). The descent of zooplankters into the hypoxic zone during the daytime
may give an energetic advantage to migrating specimens due to a decrease in the rate of oxygen consumption and locomotor
activity at low oxygen concentrations, as has been shown for females of *C. euxinus* (Svetlichny et al., 2000). This and other
experimental studies contributed to the development of an optimal behavioral strategy model (Morozov et al., 2019) for
structured populations of two species, *C. euxinus* and *P. elongatus*. The authors parameterized the model using seasonal field
observations in the NE Black Sea and showed that the diel vertical migrations of these species could be explained as the
result of a trade-off between depth-dependent metabolic costs, anoxia, available food, and predation.
Zooplankton aggregations result in sound-scattering layers (SSLs). Diel vertical migration was observed using ship
echo sounding at frequencies of 120 - 200 kHz (Erkan and Gücü, 1998; Mutlu, 2003, 2006, 2007; Stefanova and Marinova,
2015). The diurnal dynamics of *C. euxinus* and chaetognaths were documented from ship-borne echograms (Mutlu 2003,
2006). The lower boundary of the migrating *C. euxinus* was defined as $\sigma_\Theta = 16.15\text{-}16.2$ for the daytime, and the migrating
chaetognaths were defined as $\sigma_\Theta = 15.9\text{-}16.0$ (Mutlu 2007). In July 2013, a multifrequency (38, 120, and 200 kHz) ship-
borne echo-sounder survey over the southern Black Sea revealed that the daytime deep distribution of migrating *C. euxinus*
was bounded by $\sigma_\Theta$ values between 15.2 and 15.9 (Sakınan and Gücü, 2016). In the above studies, the persistent layer of
diapausing *C. euxinus* was not detected in the echograms.
The 24-h rhythm in the pattern of sound scattering was a prominent feature of the 2 MHz acoustic sensing data
obtained by a moored profiler station (Ostrovskii and Zatsepin, 2011) in the NE Black Sea. The data obtained by a short (up
to 10 days) experimental deployment of a moored automatic mobile profiler equipped with an ultrasound probe operating at
a frequency of 2 MHz and a dissolved oxygen sensor allowed Ostrovskii and Zatsepin (2011) to define the main sound-
scattering zones as follows:
- the hydrogen sulfide zone below the specific density surface $\sigma_\Theta = 15.9\text{-}16.0$ (Yakushev et al., 2005), where sound is
scattered by sedimented detritus and mineral particles, whose fluxes vary temporally while being rather homogeneous at
different depths,
- above the hydrogen sulfide zone in the suboxic layer (where the concentration of dissolved oxygen [O2] < 10 µM
(Murray et al., 1989, Oguz et al., 2006) and above that, in the oxycline ([O2] increases from 10 µM to 280-300 µM with
decreasing depth), where sound scattering occurs from both suspended particles and mesozooplankton with characteristic
sizes from 200 microns to 20 mm,
- above the oxycline in the oxygen-rich euphotic zone, where large cell phytoplankton (Yunev et al., 2020) become an
additional sound-scattering agent.
Using a combination of ultrasound sensing and stratified zooplankton sampling was necessary to resolve the ocean
fine-scale vertical distribution of mesozooplankton. An analysis of both echograms and simultaneous stratified net sampling
showed that the SSLs at 2 MHz were associated with the zooplankton species *C. euxinus* and *P. elongatus* at $\sigma_\Theta = 15.7\text{-}15.4$
and diapausing *C. euxinus* above $\sigma_\Theta = 15.9$ (Arashkevich et al., 2013).
The specific theme of this study is the seasonal change in the sound-scattering zooplankton vertical distribution across
the oxygen gradient from the lower part of oxygenated water to the anoxic zone boundary. This theme is in line with the EU
Horizon 2020 BRIDGE-BS project (http://www.unsdsn.gr/h2020-bridge-bs), which focuses on Black Sea ecosystem
functioning. While the project relies on future observations and methods for understanding biogeochemical processes at
several pilot sites, this paper presents ongoing observations at the northeastern Black Sea Gelendzhik site. The acoustic data
were collected year round and analyzed to infer the SSL seasonal variability in relation to the oxygen stratification. Our
observational study was made possible using a moored Aqualog profiler equipped with an ultrasound probe, a CTD probe,
and a fast oxygen sensor. The advantage of this approach is that it provides frequent year-round measurements (with an
interval of up to 1 h) of collocated vertical profiles of sound scattering, temperature, salinity, and dissolved oxygen
concentration in the water column from the near-surface to the bottom layer with a high vertical resolution (up to 20 cm).
This helps to fill in the gaps due to insufficient zooplankton sampling in the winter season and resolves difficulties with
sampling at precise depths, thereby providing the information needed to define the displacements of the mesozooplankton
aggregations.
The goals of the analysis are as follows: (1) to develop methods to visualize the SSLs in the lower part of the oxycline
and in the hypoxic zone, (2) to validate the SSLs in the oxygen-deficient waters using the taxonomic and quantitative
characteristics of zooplankton vertical distribution derived from stratified net sampling, and (3) to describe the seasonal
variations of the deep mesozooplankton SSLs, including the diapause duration of CV *C. euxinus*, in relation to oxygen
concentration (the oxygen bounds for the mesozooplankton SSLs).
**2 Measurements**
This study is based on the comparative analysis of the amplitude of sound backscattering data at a frequency of 2 MHz and
oxygen concentration data in seawater obtained in the NE Black Sea using a moored automatic mobile profiler Aqualog (Fig.
1) (Ostrovskii and Zatsepin, 2011, 2016; Ostrovsky et al., 2013). To obtain the depth profiles of the volume backscattering
strength, the Aqualog profiler was equipped with a Nortek Aquadopp acoustic Doppler current meter
(https://www.nortekgroup.com/assets/documents/ComprehensiveManual_Oct2017_compressed.pdf).

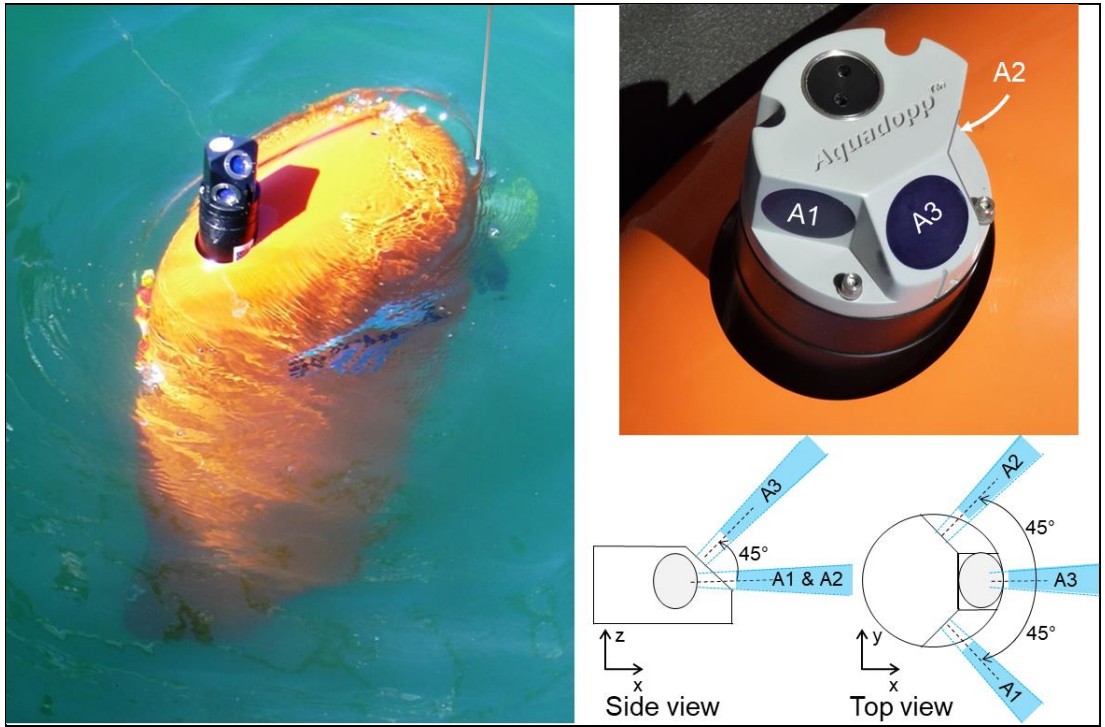

**Figure 1: The moored Aqualog automatic mobile profiler with a deep-water Aquadopp acoustic Doppler current meter (left). Transducer head of the shallow-water Aquadopp acoustic Doppler current meter on the Aqualog profiler (right). Bottom right: the acoustic beams are shown in blue and are labeled $A_1$, $A_2$, and $A_3$.**

The Aquadopp is a narrow-band instrument (https://support.nortekgroup.com/hc/en-us/articles/360029839331-The-
Comprehensive-Manual-ADCP) that emits short sound pulses (pings) at a constant frequency and receives reflected (echo)
signals. Plankton and suspended matter, as well as air and gas bubbles, are the main scatterers of the sound. While sound
pulses are scattered in all directions when they hit particles, a small fraction of the incident sound pulse intensity is reflected.
The Aquadopp current meter employs a mono-static system in which 3 transducers are used to transmit and receive signals at
an acoustic frequency of 2 MHz. Measurements are made in the 90 dB range with a resolution of 0.45 dB. In high-accuracy
acoustic Doppler measurements, the acoustic beams are narrow and each has a cone angle of 1.7°. Three focused beams
measure the scattering strength with high sampling rates in a small volume (referred to as a single point). Two-sided acoustic
beams are directed horizontally with 90° spacing between the axes of the beams (Fig. 1). These beams measure the volume
scattering strength at the level of the transducer. The third beam is inclined at an angle of 45° to the plane formed by the axes
of the other two beams. The piezoelectric element of the transducer transmits sound waves when it vibrates. The vibration
does not stop at once but is damped over time. The speed of sound in water and the damping time of the membrane vibration
determine the dead zone. In our case, this distance along the acoustic beam is approximately 0.35 m from the piezoelectric
element of the transducer to the measurement cell (in the form of a truncated cone). The sound pulses are scattered and
reflected back to the transducer. In our case, the length of the cell along the axis of the acoustic beam is approximately 1.5
m. Therefore, the reflected sound pulse intensity obtained by the instrument is the weight average for the time during which
the sound wave passes the distance of 1.85 m to the far boundary of the measurement cell plus 1.85 m on the way back. The
received signal is processed in such a way that the greatest contribution to the average value is made by the scattering in the
center of the measurement cell at a distance of approximately 1.1 m from the transducer. The device can transmit up to 23
sound pulses every 1 s. The average value of the volume scattering strength for sound pulses transmitted and received in 1 s
is recorded in the device's memory.
The high frequency of 2 MHz allows observations of small-sized sound scatterers. Theoretically, a 2 MHz transducer
is most sensitive to particles with a diameter of 0.23 mm (estimates for different frequencies for standard seawater are given,
for example, in Hofmann and Peeters, (2013)). However, this is not entirely applicable to zooplankton due to the complex
shape of these organisms, their structure, their lipid composition, and the presence of gases in their bodies (Stanton et al.,
1994; Lavery et al., 2007, Lawson et al., 2006). However, as a simplified model, copepod species are often considered
cylinders, the scattering from which is defined as a function of the incident sound pressure, the acoustic wavelength, and the
distance between the transmitter and the animal. An approximate formula for describing sound scattering from an elongated
weakly scattering body of an animal also includes the angle of orientation of the body (Stanton et al., 1993, 1994).
Unfortunately, the manufacturer of the Aquadopp instrument does not specify information about the acoustic power of its
transducers. The Aquadopp measurement data for the volume scattering strength are presented in conventional units
(counts). Without special calibration, it is not possible to determine the amount of falling sound pressure in water at a
distance from the instrument transducer.
Since 2013, Aquadopp instruments with sideways-looking vertically mounted heads have been regularly used on the
Aqualog profiling carrier (Fig. 1). The carrier moves up or down at a speed of approximately 0.2 m s$^{-1}$, so the vertical
resolution of the volume scattering strength data is 0.2 m. These data are averaged every 5 s, allowing for the detection of an
SSL with a thickness on the order of 1 m.
In the context of this study, the ability to observe sound that has been reflected from zooplankton species at different
angles is important. In the case of settling detritus, the volume scattering strength of slanted beam $A_3$ and those of horizontal
beams $A_1$ or $A_2$ are approximately the same. If the elongated suspended particles are oriented vertically or inclined, the
amplitude of $A_3$ will significantly differ from the amplitudes of $A_1$ and $A_2$. This was shown for copepods based on both
models of acoustic scattering at a frequency of 2 MHz (Stanton and Chu, 2000; Roberts and Jaffe, 2007) and laboratory
experiments (Roberts and Jaffe, 2008).
Thus, by comparing the amplitudes $A_1$, $A_2$, and $A_3$, one can judge the predominant orientation of species in
zooplankton aggregations. It is assumed that aggregation's characteristic size is greater than the length of the acoustic
measurement cell, that is, not less than ~2 m, and its lifetime is longer than 10 s. Therefore, during the Aqualog carrier
movement at a speed of 0.2 m s$^{-1}$, the slanted and horizontal acoustic beams scan the same zooplankton aggregation. The
complexity and variability of the acoustic backscatter makes it difficult to compare the acoustic signals obtained for different
observational periods. Proper normalization of the signals is needed to evaluate the seasonal change in the vertical
distribution of the mesozooplankton SSLs from many profiles despite the variability of the amplitude of the acoustic
backscatter. For the Aquadopp instrument, such normalization is the ratio of the volume scattering strength of the horizontal
beams to the volume scattering strength of the slanted beam
$$R = (A_1 + A_2)/2A_3. \tag{1}$$
It allows for a drastic reduction in the noise associated with clouds of sinking particles, which have an approximately equal
area in the horizontal projection to the projection with a 45° angle of inclination. In some cases, the suspended particles can
completely obscure the signal associated with the aggregation of mesozooplankton. However, in this study, there were
usually only a few such cases. As will be shown below in Section 3, typically at depths from 60 to 120 m during the day,
the directional acoustic backscatter ratio $R = 1.05$-$1.2$, and at night, $R < 1.05$. In Appendix, we will consider whether the
mesozooplankton specimens' vertical orientation is tilted in the deep aggregations. The analysis will be based on calculation
of the ratio of the volume scattering strength of the horizontal beams $A_1/A_2$ assuming that due to the tilt the standard
deviation of $A_1/A_2$ should be greater than 0.

163       In addition to the Aquadopp instrument, a SeaBird 52MP CTD probe and Aanderaa 4330F and SBE 43F dissolved

oxygen fast sensors were incorporated into the Aqualog profiler aerobic zone (Ostrovskii and Zatsepin, 2016). The SeaBird
52MP CTD was specially designed for a moored profiling application in which the instrument makes vertical profile
measurements from a carrier that travels vertically beneath a subsurface floatation (https://www.seabird.com/moving-
platform/sbe-52-mp-moored-profiler-ctd-optional-do-sensor/family?productCategoryId=54627473795). The CTD is
equipped with a pump that controls a flow at a constant speed through a single small diameter opening to ensure the
minimization of salinity spiking in the measurement data by the temperature and conductivity cell. On the Aqualog profiling
carrier slowly moving at ~0.2 m s$^{-1}$, the CTD sampling rate of once per second provides sufficient data to resolve ocean fine-
scale thermohaline structure. The accuracy of the CTD probe is 0.002 °C for the temperature, ± 0.0003 S/m for the
conductivity and ± 0.1% of the full scale range for the pressure. The SBE 43F accuracy should be no worse than ±2%
saturation, which can be compared with 5% for Aanderaa 4330F with a resolution better than 1 µM or 0.4%
(https://www.aanderaa.com/media/pdfs/d378_aanderaa_oxygen_sensor_4330_4330f.pdf). In practice, in the Black Sea, SBE
43F showed very robust results in detecting the lower boundary of the oxic zone, consistent with observations of the sigma-
density structure and definition of the oxic zone boundary for the northeastern region of the Sea (Ostrovskii and Zatsepin,
2016). The SeaBird 52MP CTD with SBE 43F was regularly calibrated at the facility of the Southern Branch of Shirshov
Institute of Oceanology, Gelendzhik. The dissolved oxygen measurements by using the Aanderaa 4330F and SBE 43F
sensors at the profiler were described in (Ostrovskii and Zatsepin, 2016) and later in a companion paper (Ostrovskii et al.,
2018). The fast response sensing foils of the Aanderaa 4330F sensor were replaced by new foils two times in the past four
years. The CTD and dissolved oxygen sensors were mounted at the leading edge of the Aqualog profiler pointing into
horizontal oncoming flow, while hydrodynamic cowling (vertically oriented, wing-like) helped to stabilize the profiler
orientation with respect to the flow direction. It should be noted that the Black Sea environment is particularly suitable for
profiling measurements since there is no biological fouling on the sensors of the profiler, which is usually submerged into
the hydrogen sulfide zone for ~10 min every 1-2 h. Finally, the dissolved oxygen sensor data were verified with the water
samples at standard depths for determination of dissolved oxygen by Winkler method (not shown here).
The profiler mooring station was deployed approximately four nautical miles from the coast at the uppermost part
of the continental slope at 44°29.3′N and 37°58.7′E (Fig. 2). From June 2010 to April 2021, 16 surveys lasting from a few
days to 3 months were carried out (Table 1) (Solovyev e al., 2021). During the surveys, the device automatically performed a
profiling cycle usually every 1-2 h, descending to the near-bottom depth of 200-220 m and ascending to the upper layer
while remaining submerged at a depth of 20-40 m. In particular, in 2016-2020, more than 14,000 multiparameter sets of
vertical profiles were collected year-round (except March).

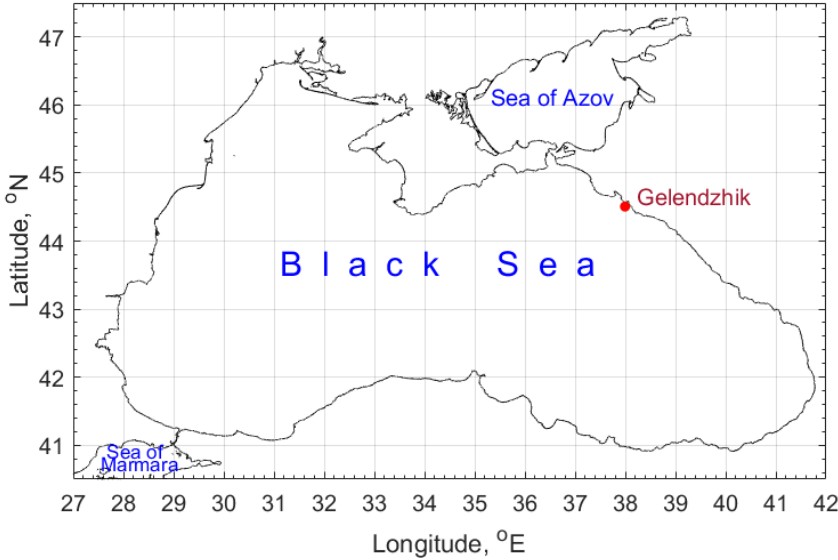


**Figure 2: The Black Sea coastline (http://openstreetmapdata.com/data/coastlines). The observational site off Gelendzhik is shown**
**by a red dot. © OpenStreetMap contributors 2021. Distributed under the Open Data Commons Open Database License (ODbL)**
**v1.0.**
To acquire taxonomic and quantitative features of zooplankton vertical distribution, stratified net samples were
taken from R/V Ashamba (Table 1) near the moored profiler Aqualog with a Juday net (mouth area 0.1 m$^2$, mesh size 180
μm) equipped with a closing device. The towing speed was 0.9-1.0 m s$^{-1,}$ and the net was closed without stopping the upward
movement. The sampling was carried out in calm weather so that the wire angle was not higher than 10 degrees. The
sampling was carried out at earlier stages of this project in June 2010, October 2013, and July 2014, as well as later in
October 2016.
**Table 1. Deployments of the profiler Aqualog-6 with Nortek Aquadopp current meter in the NE Black Sea and the dates**
**of the zooplankton sampling near the profiler mooring site in 2010-2021. Since 2013, the profiler Aqualog was equipped with SBE**
**52MP CTD probe with SBE 43F DO sensor. Additional sensors used at the profiler were as follows Oxygen Aanderaa 4330F,**
**Seapoint Turbidimeter, and Seapoint Fluorometer.**

| Sur-vey | Start | End | Profile cycle interval, h | Profile depth range, m | Number of profiles | Additional sensors at the profiler | Stratified net sampling for zooplankton / Sampling for determination of dissolved oxygen by Winkler method |
|---|---|---|---|---|---|---|---|
| 1 | 21.06.2010 16:03 | 22.06.2010 16:50 | 1 | 19-245 | 25 | *No dissolved oxygen sensor | Zooplankton: 21.06.2010 18:05-19:00 21.06.2010 21:10-21:55 22.06.2010 00:05-00:50 22.06.2010 05:30-06:20 22.06.2010 09:00-09:50 |
| 2 | 02.10.2013 12:42 | 07.10.2013 9:14 | 1 | 30-220 | 234 | -- | Zooplankton: 06.10.2013 12:30-13:20 |
| 3 | 28.06.2014 10:46 | 02.07.2014 13:24 | 1 | 20-240 | 198 | 4330F | Zooplankton: 01.07.2014 13:30-14:30 02.07.2014 02:30-03:30 Dissolved oxygen: 12.07.2020, 14.07.2020 16.07.2020 |
| 4 | 06.10.2014 5:50 | 17.12.2014 12:02 | 6 | 30-220 | 860 | 4330F | -- |
| 5 | 01.01.2016 18:00 | 06.03.2016 6:00 | 2 | 28-208 | 1490 | 4330F | -- |
| 6 | 06.10.2016 5:47 | 10.10.2016 10:21 | 2 | 25-220 | 98 | 4330F, Fluorometer, Turbidimeter | Zooplankton: 04.10.2016 22:00-23:00 05.10.2016 11:05-11:50 |
| 7 | 10.10.2016 13:24 | 12.11.2016 12:45 | 2 | 30-220 | 790 | 4330F, Fluorometer, Turbidimeter | -- |
| 8 | 10.02.2019 12:00 | 24.02.2019 4:08 | 2 | 25-206 | 328 | Turbidimeter | -- |
| 9 | 16.04.2019 11:34 | 28.05.2019 9:24 | 1 | 46-206 | 2016 | 4330F, Turbidimeter, | -- |
| 10 | 01.06.19 10:32 | 27.08.19 12:02 | 1-2 (16 cpd) | 22-200 | 2784 | Turbidimeter | Dissolved oxygen: 06.07.2019, 08.07.2019 12.07.2019 |
| 11 | 30.08.19 16:00 | 15.10.19 20:26 | 1-2 (16 cpd) | 22-200 | 1482 | Turbidimeter | -- |
| 12 | 28.10.19 14:00 | 24.12.2019 20:36 | 1-2 (16 cpd) | 21-204 | 1491 | 4330F | -- |
| 13 | 28.03.2020 11:30 | 24.05.2020 2:03 | 1-2 (16 cpd) | 20-200 | 1584 | 4330F, Fluorometer | -- |
| 14 | 16.07.2020 5:00 | 26.07.2020 23:13 | 1-2 (16 cpd) | 23-201 | 444 | 4330F, Fluorometer | Dissolved oxygen: 17.07.2020, 20.07.2020 |
| 15 | 03.10.2020 5:00 | 27.11.2020 9:37 | 2 | 20-203 | 1320 | 4330F, Fluorometer | -- |
| 16 | 11.12.2020 9:06 | 07.04.2021 1:04 | 4 | 21-203 | 1399 | 4330F, Fluorometer **Nortek Aquadopp broken | -- |


The net hauls targeted the backscattering aggregation considering that their locations were associated with specific
isopycnal layers (Ostrovskii and Zatsepin, 2011, Fig. 9). Vertical profiles of temperature, salinity, and density were obtained
with a ship-borne SeaBird 19plus CTD probe prior to mesozooplankton sampling. Depth strata were chosen based on the
CTD profiles to sample the upper mixed layer (UML), the thermocline layer, the layer from the oxycline upper boundary ($\sigma_\Theta$
= 14.25) to the lower boundary of the thermocline, and two layers in the oxygen-deficient zone: the layer from depths of $\sigma_\Theta$
15.7 to $\sigma_\Theta$ 15.4 and the layer from 2-3 m below $\sigma_\Theta$ 15.9 to $\sigma_\Theta$ 15.7.
The time of sampling corresponded to the day/night vertical distribution and upward/downward migration of
zooplankton (June 2010), the daytime distribution (October 2013), and the day/night distribution (July 2014 and October
2016). The samples were immediately fixed with buffered formaldehyde (4% final concentration of seawater–formaldehyde
solution). The volume of filtered sea water was estimated from the area of the net mouth and the length of the released wire.
Organisms were identified and counted under a stereomicroscope equipped with an ocular micrometer. Zooplankters were
identified at the level of species and age stages of copepods and size classes (with an interval of 2 mm) of chaetognaths and
ctenophores. The smallest organisms (meroplankton, appendicularians, copepod nauplii and ova) considered in the analysis
were 180 μm in size. Mesozooplankton biomass in terms of dry weight (DW) was estimated based on the published length-
DW regressions for different species summarized in (Arashkevich et al., 2014, Table 2). Biomass values were standardized
to mg DW $m^{-3}$ or mg DW $m^{-2}$. The intensity of the echo signal strongly depends on the material properties of the organism's
tissue (Stanton et al., 1994); therefore, when comparing the pattern of the scattering signal intensity with the pattern of
zooplankton distribution in a community containing different taxa, it was reasonable to express zooplankton biomass as DW
or carbon (Flagg and Smith, 1989; Heywood et al., 1991; Ashjian et al., 1998). For a graphical presentation of the results, six
components of zooplankton were considered: copepods *Calanus euxinus* and *Pseudocalanus elongatus*, small crustaceans
(*Acartia clausi*, *Paracalanus parvus*, *Oithona similis* and cladocerans), heterotrophic dinoflagellate *Noctiluca scintillans*,
chaetognaths *Parasagitta setosa*, and varia (ctenophores *Pleurobrachia pileus*, appendicularians, meroplankton, decapod
larvae, Pisces ova).
One method for calculating vertical migration speed of zooplankton from the sound backscatter data of the acoustic
current meter at the profiler Aqualog was described in (Pezacki et al., 2017). However the vertical migration speed of
mesozooplankton is beyond the focus of this study. Only once when discussing the pattern of the diel vertical migration, the
slope of the migration track on the echogram (see Figure 9 below) is considered to give rough idea about the dive and the
ascend of  mesozooplnakton. Much more effort would certainly be needed to visualize the specimens' vertical swimming.

## 3 Results

### 3.1 Acoustic scattering by mesozooplankton aggregations

The first validation data for the Aquadopp observations were obtained on 21-22 June 2010. The sound scattering layers were identified at the raw echogram (Fig. 3) as mesozooplankton aggregations by comparison with the net sampling data (Fig. 4). The zooplankton net sampling data were consistent with the acoustic backscatter, indicating short-term variations in biomass and diel vertical migration of zooplankton.

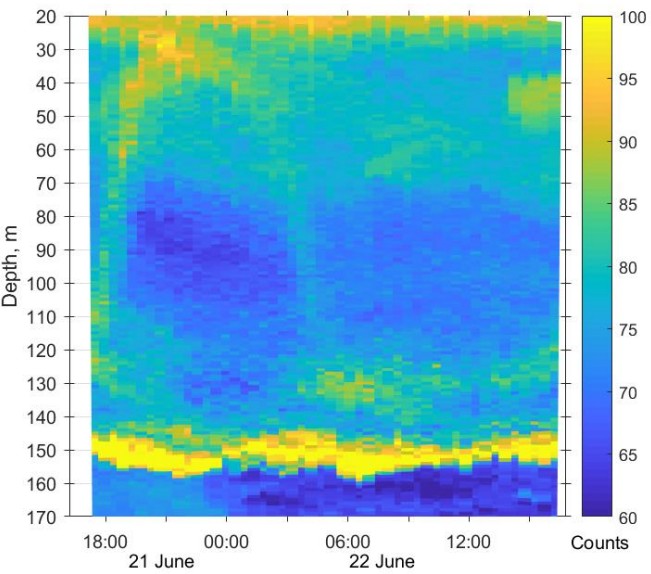

**Figure 3: Diurnal motions of the sound scattering layers in the oxic zone. The depth-time scatterplot for profiles of acoustic backscatter amplitude at 2 MHz was obtained using the Aquadopp instrument at the moored profiler during verification study involving net sampling of zooplankton on 21-22 June 2010.**

The total mesozooplankton biomass in the entire water column varied from 0.99 to 3.57 g DW m$^{-2}$. Zooplankton was dominated by the copepod *Calanus euxinus*, which made up the mean 58% with the standard deviation (SD) ±14% of the total biomass (below for the sake of brevity, such estimates are denoted as mean±SD%). The contribution of chaetognaths *Parasagitta setosa* was 21±11%, followed by copepods *Pseudocalanus elongatus* (13±7% of the total biomass). The sum share of other groups of mesozooplankton did not exceed 7% of the total biomass.

The pattern of the vertical distribution of mesozooplankton biomass reveals a relatively uniform distribution over depth in the evening twilight (18:05-19:00) and at dawn (05:30-06:20) (Fig. 4, left column). At night (21:10-21:55 and 00:05-00:50), the highest concentration of zooplankton was observed in the thermocline layer, while in the daytime (09:00-09:50), the zooplankton maximum was in the layer between the density surfaces $\sigma_\Theta$ 15.7 and 15.4 (Fig. 4, left), in accordance with the diurnal changes in the volume backscatter strength (Fig. 3). The deepest layer bounded by isopycnals $\sigma_\Theta$ 15.9 and 15.7 was inhabited by nonmigrating copepods, the fifth copepodite stage (CV) of *C. euxinus* (median prosome length 2.3

mm), persistently staying at this depth throughout the day (Figs. 3 and 4, middle column). Visual inspection of live samples
revealed quiescent behavior of these specimens and large oil sac volume inside their body, suggesting a diapausing state in
*C. euxinus* CV collected from the deepest layer (Vinogradov et al., 1992).

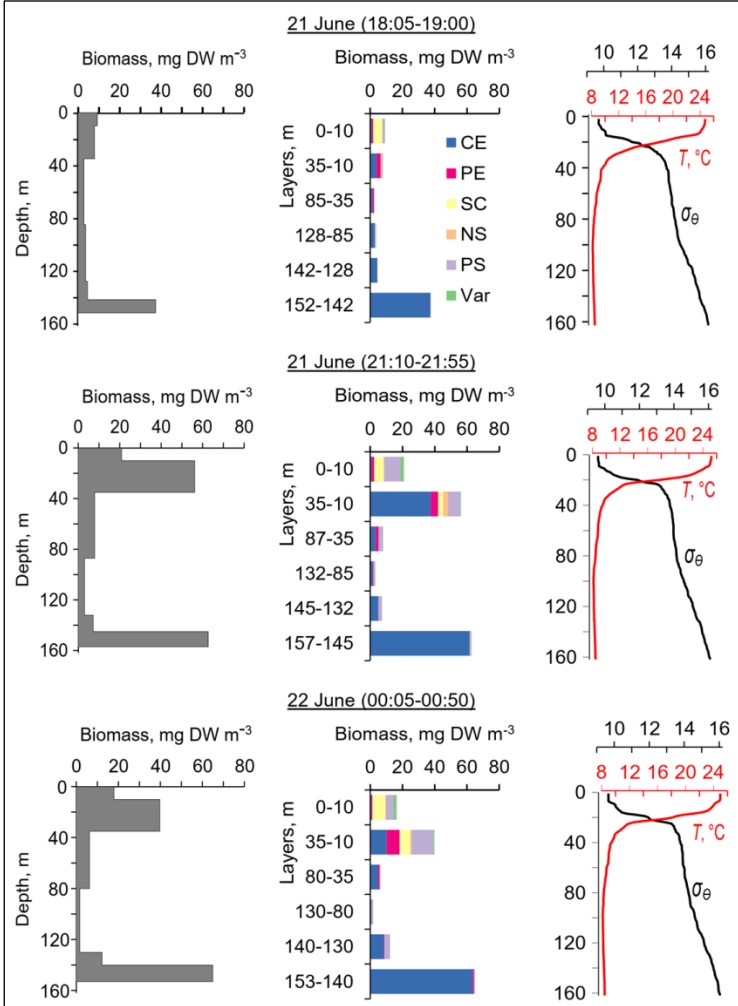


**Figure 4: The diel changes in vertical distributions of (left) total mesozooplankton biomass, (middle) zooplankton composition,**
**(right) temperature (*T*) and density ($\sigma_\theta$) near the mooring site on 21-22 June 2010. The temperature and density profiles were used**
**for the selection of sampling strata. CE – *Calanus euxinus*; PE – *Pseudocalanus elongatus*; SC – small crustaceans; NS – *Noctiluca***
***scintillans*; PE – *Parasagitta setosa*; Var – varia.**

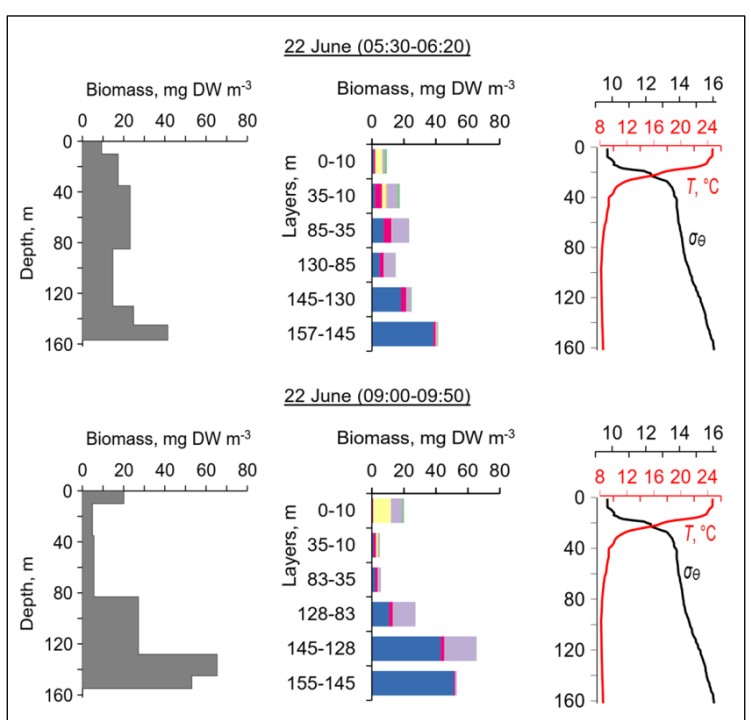

**Figure 4: Continued.**

Three migrating species, copepodites CIV-CVI of *C. euxinus* (median prosome length 2.6 mm), CV-CVI *P. elongatus* (median prosome length 0.92 mm), and chaetognaths *P. setosa* (median length 19 mm), formed daytime zooplankton aggregations in the oxygen-deficient zone (Fig. 4, middle panel). Ctenophore *Pleurobrachia pileus*, also inhabiting the deep layers in the daytime, contributed negligibly to the total biomass due to the low dry matter content in their gelatinous bodies and their low abundance (shown as Var in Fig. 4). At night, most of the migrating zooplankters were concentrated in the thermocline and did not ascend to the warm UML, which was inhabited by small copepods, cladocerans, and small (<6 mm) chaetognaths.

## 3.2 Zooplankton aggregations visualized using the directional acoustic backscatter ratio

The echograms based on the data from horizontal-beam transducers $A_1$ and $A_2$ often reveal aggregations of zooplankton at depths of 80-120 m in the daytime (Fig. 5). The aggregations begin to rise around sunset and descend before dawn. Thin, nearly vertical lines on the echogram indicate acoustic traces of the migrating mesozooplankton species. The echogram also shows patches that occupy the entire water column, from the upper to the lower measurement depth, penetrating below the surface $\sigma_\Theta$ 15.9 and then deeper into the hydrogen sulfide zone. These are clouds of suspended particles (see, for example, Klyuvitkin et al., 2016). Acoustic scattering by clouds of particles sinking through the water column can obscure zooplankton aggregations.

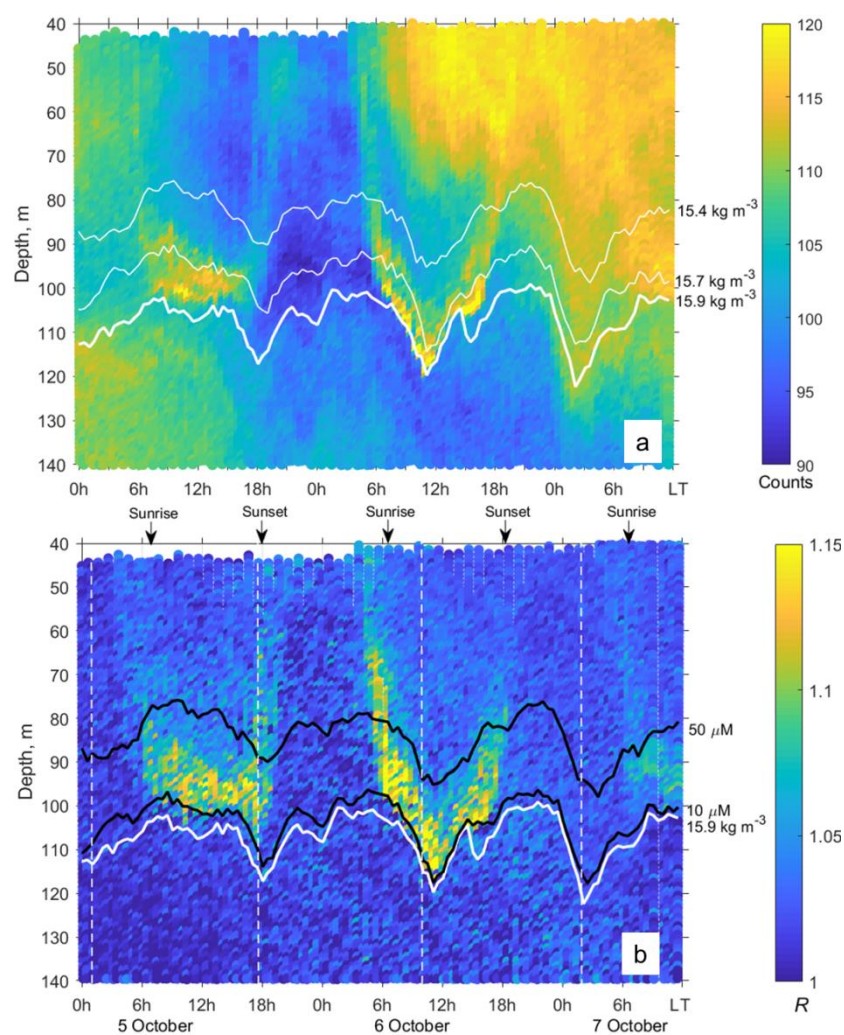

**Figure 5: a - The time-depth graph of the Aquadopp horizontal-beam $(A_1 + A_2)/2$ echogram showing acoustic backscatter intensity**
**(counts) on 5-7 October 2013. The Aqualog profiler with the Aquadopp instrument performed ascending/descending cycles every 1**
**h. The upper, middle and lower white lines are for isopycnals $\sigma_\Theta$ 15.4, 15.7 and 15.9, respectively. b – Time-depth scatterplot of the**
**Aquadopp directional acoustic backscatter ratio $R = (A_1 + A_2)/2A_3$. Colored lines show [O2] = 50 μM (upper black line); [O2] = 10**
**μM (lower black line); and $\sigma_\Theta$ = 15.9 kg m$^{-3}$ (white line), which can be taken as a proxy for the boundary of the oxygen zone in the**
**NE Black Sea (Glazer et al., 2006, Ostrovskii and Zatsepin, 2016). Notice that due to upwelling the oxycline was moved upward.**
**Vertical dotted white lines indicate a 17.3 h time interval, which is equal to the period of inertial oscillations at the latitude of the**
**observation. They approximately coincide with troughs of inertial waves.**

The layers of elevated acoustic backscatter amplitude due to deep zooplankton aggregations are accounted for using
the $R$ graphs (Fig. 5b) that were validated by net sampling on 6 October 2013 (Fig. 6), although sampling was not performed
at night due to stormy weather. Since the depths of the isopycnals of 15.9 and 15.7 differed by only 3 m, the integrated
zooplankton sample was taken in the layer between $\sigma_\Theta$ = 15.9 and $\sigma_\Theta$ = 15.4. In this layer, the contributions of *Calanus*
*euxinus*, *Parasagitta setosa*, and *Pseudocalanus elongatus* to the total biomass were 60%, 26%, and 12%, respectively (Fig.
6 b). The extremely low zooplankton biomass (< 2 mg DW m$^{-3}$) in the upper 50 m layer (Fig. 6 a, b) is consistent with data
on a fourfold decrease in the annual average biomass of upper-dwelling zooplankton in 2013 compared to previous years
(Arashkevich et al., 2015).

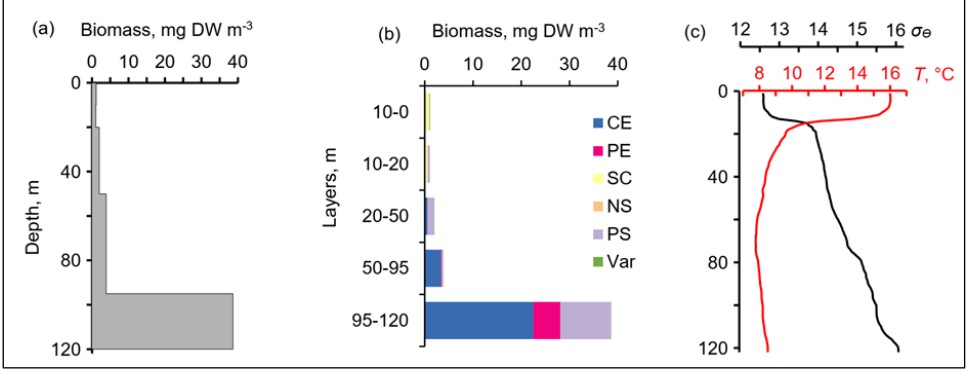

**Figure 6: The daytime vertical distributions of (a) total mesozooplankton biomass, (b) zooplankton composition, (c) temperature**
**(*T*) and density ($\sigma_\Theta$) near the mooring site at 12:30-13:20 on 6 October 2013. Temperature and density profiles (c) indicate the**
**selection of sampling strata. CE – *Calanus euxinus*; PE – *Pseudocalanus elongatus*; SC – small crustaceans; NS – *Noctiluca***
***scintillans*; PE – *Parasagitta setosa*; Var – varia.**
Zooplankton diel vertical migration trajectories in the *R* graph are noticeably clear below 40 m (Fig. 5b). The
explanation of these phenomena could be that the scattering area of the elongated bodies of the zooplankton species is larger
in the horizontal projection than in the inclined projection at an angle of 45°. Therefore, the orientation of the bodies of
mesozooplankton species appears to be mainly vertical during migration. At night, these specimens are randomly oriented in
the upper layer, where $R \approx 1$. In addition to the diel vertical migrations, intraday vertical fluctuations of zooplankton occur
with an inertial period (Fig. 5b). The vertical displacements of the daytime deep mesozooplankton aggregations are coherent
with the vertical displacements of both isopycnals and isooxylines. The displacements of isopycnals with amplitudes up to
20 m are mainly due to near-inertial waves.
In October 2016, persistent aggregation of diapausing *C. euxinus* was detected in the acoustic backscatter signal (Fig.
7), unlike October 2013 (Fig. 5). Zooplankton sampling was performed at midnight and midday on 4-5 October 2016 (Fig.
8). The pattern of zooplankton distribution was similar to that in June 2010 (Fig. 4), both in terms of the total biomass and
composition of zooplankton and in terms of the day/night vertical distribution. The total mesozooplankton biomass of 1.8-
2.3 g DW m$^{-2}$ was dominated by three species: *C. euxinus* (59-76%), *P. setosa* (9-22%), and *P. elongatus* (5-10%). At night,
the maximum aggregation of migrating zooplankters was in the thermocline layer, and at midday, it was in the layer between
isopycnals $\sigma_\Theta$ 15.7 and 15.4 (Fig. 8a). Daytime zooplankton aggregation consisted mainly of *C. euxinus* (92% of total
biomass) with a small contribution from chaetognaths (7% of total biomass) (Fig. 5b). The layer between isopycnals $\sigma_\Theta$ 15.9
and 15.7 was persistently occupied by diapausing *C. euxinus* CVs. The chaetognaths found in this layer were represented by
spent specimens and corpses (Fig. 8b). UML was inhabited by nonmigrating small copepods, cladocerans, and small
chaetognaths.

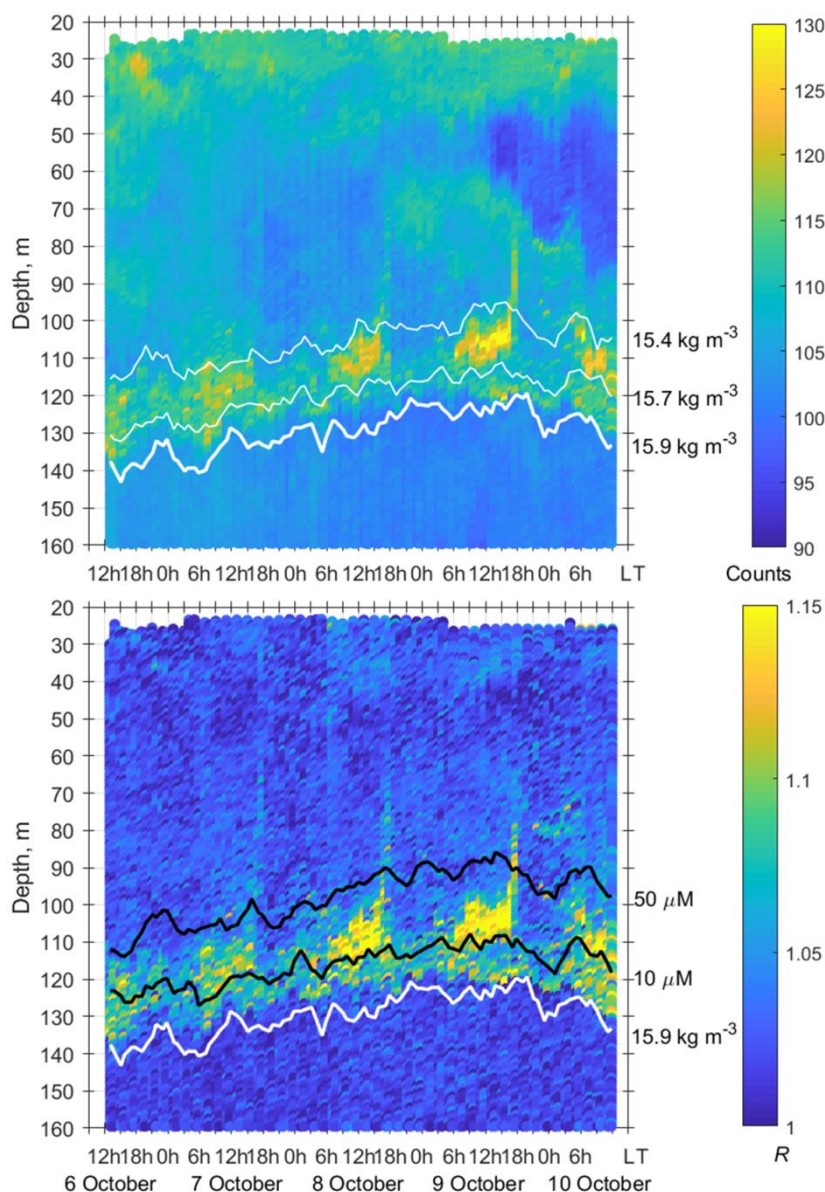


**Figure 7: a - The time-depth scatterplot of the Aquadopp horizontal-beam echo (A1 + A2)/2 on 6-10 October 2016. b – The time-depth graph of the Aquadopp directional acoustic backscatter ratio *R*. The isopycnals and isooxylines are superimposed near the SSLs.**


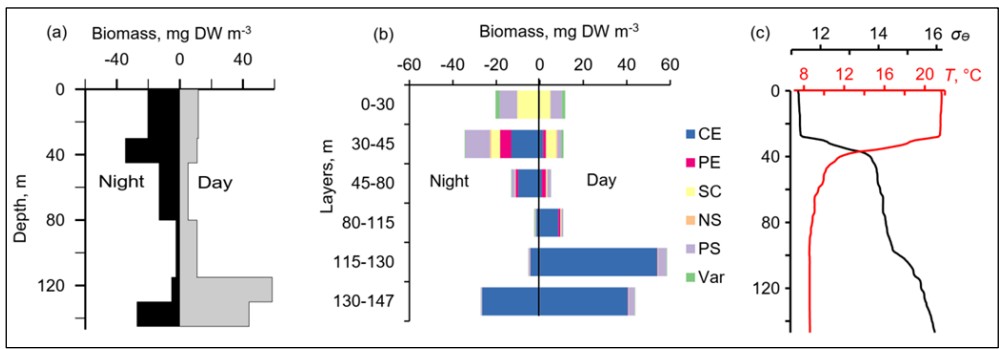

**Figure 8: Day/night vertical distribution of (a) total mesozooplankton biomass, (b) zooplankton composition, (c) temperature ($T$, °C) and density ($\sigma_\theta$) near the mooring site at 22:00-23:00 of 4 October (night) and 11:05-11:50 of 5 October (day) in 2016. Temperature and density profiles (c) indicate the selection of sampling strata. CE –** *Calanus euxinus***; PE –** *Pseudocalanus elongatus***; SC – small crustaceans; NS –** *Noctiluca scintillans***; PE –** *Parasagitta setosa***; Var – varia.**

The net sampling data on the day/night vertical distribution of mesozooplankton agreed broadly with the acoustic backscatter observations obtained during the next few days (Fig. 7). On the echogram, one can see a persistently existing backscattering layer associated with the isopycnal layer near $\sigma_\Theta = 15.9$ as well as patches of the high-volume backscattering strength at depth during the daytime and their movement into shallower layers at night.

The two-layered structure was also observed at the end of June – early July 2014 (Fig. 9) and validated by day/night zooplankton sampling on 1-2 July (Fig. 10). Deeper zooplankton aggregation was monospecific, consisting only of diapausing *C. euxinus* CVs (Fig. 10b) and formed a thin layer (5 - 10 m thick). This layer was visible all day and night and was usually located above the isopycnal surface of 15.9. It is clearly distinguished by the value $R > 1.1$ (Fig. 9b). The daytime zooplankton aggregation consisted of three migrating species and their different developmental stages, *C. euxinus*, CIV-CVIs, *P. elongatus*, CV-CVIs, and *P. setosa*, 14-22 mm in size (Fig. 10b). Since the amplitude of vertical migration is different for different components of this assembly, the daytime deep aggregation reached 35 m in thickness. Before sunset, migrating zooplankters began to move upward and at night formed aggregations at depths above 40 m (Fig. 9a) and peaked in the thermocline at 17-25 m (Fig. 10a).

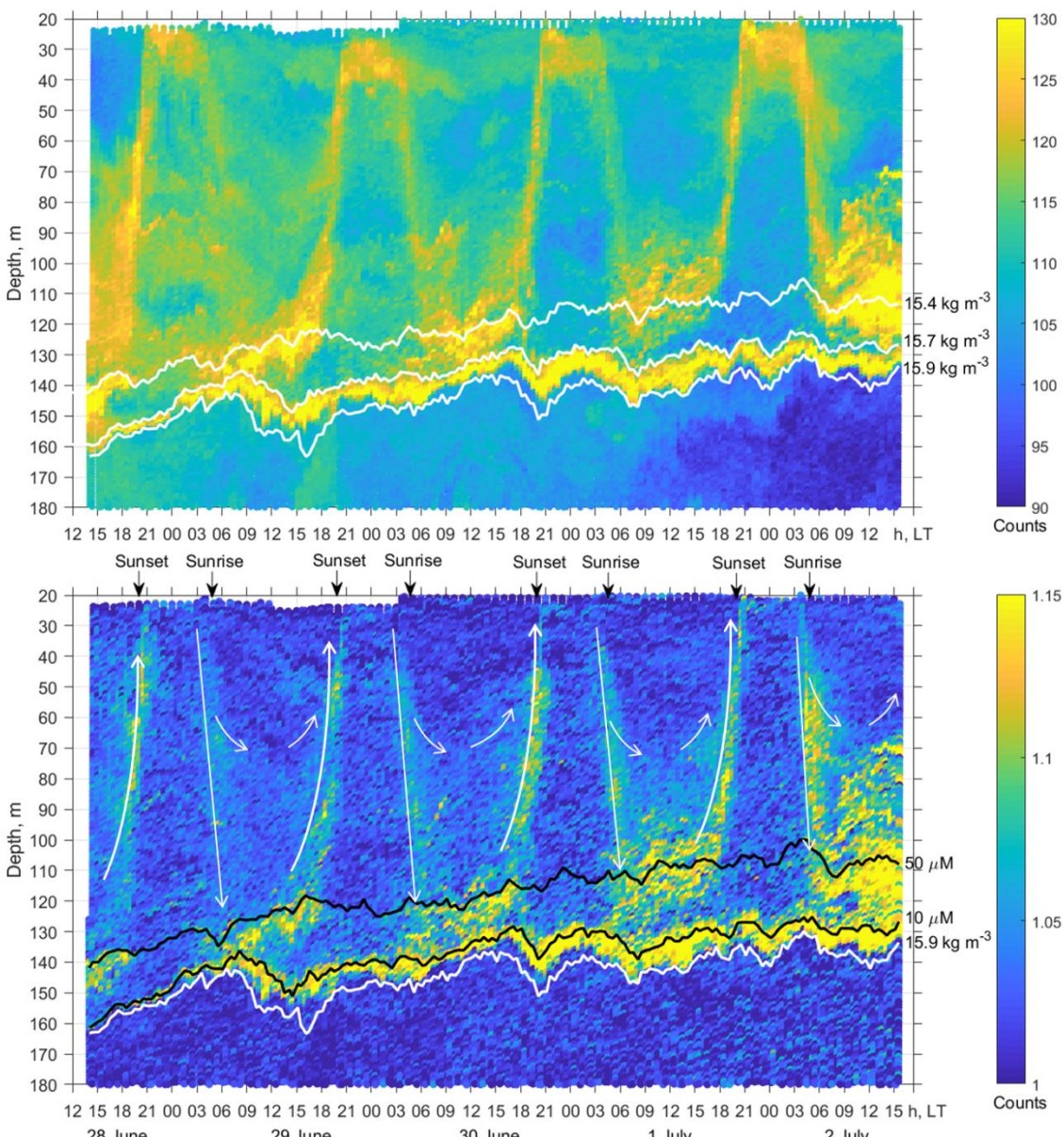

**Figure 9: a -– The time-depth graph of the Aquadopp horizontal-beam echo (A1 + A2)/2 during the moored profiler survey on 28**
**June - 2 July 2014. The isopycnals are superimposed near the SSLs. b - The graph of time-depth variation in _R_ based on the**
**measurements of sound backscattering. The upper and lower black lines are iso-oxylines of 50 and 10 μM, respectively. The white**
**line indicates isopycnal $\sigma_\Theta$ = 15.9. There is a persistent SSL under isooxyline [O2] = 10 μM. Thin white arrows schematically show**
**the diel migration of mesozooplankton. The maximum depth of the diel vertical migration is 120-150 m, although some specimens**
**dive to depths of only 80-100 m. The slope of the straight arrow pointing downwards corresponds to a diving speed of ~1.5 cm s$^{-1}$.**
**The ascent is accelerated and reaches values of approximately 2.5 cm s$^{-1}$ in the upper 60 m depth.**


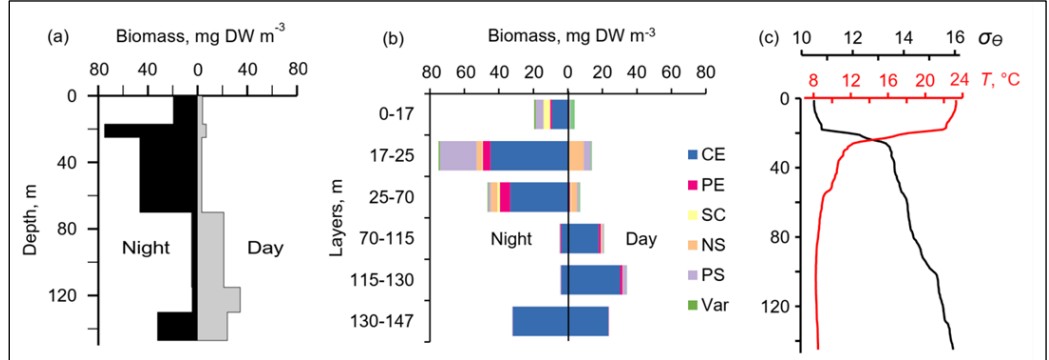


**Figure 10: The day/night vertical distribution of (a) total mesozooplankton biomass, (b) zooplankton composition, (c) temperature (T) and density ($\sigma_\theta$) near the mooring site on 1 July (13:30-14:30) and 2 July (02:30-03:30) of 2014. Temperature and density profiles (c) indicate the selection of sampling strata. CE – *Calanus euxinus*; PE – *Pseudocalanus elongatus*; SC – small crustaceans; NS – *Noctiluca scintillans*; PE – *Parasagitta setosa*; Var – varia.**

Since migrating zooplankton aggregations were observed in the deep layers only during the daytime, it is worth comparing the daytime average *R* profile with that for the nighttime (Fig. 11). Such a comparison clearly reveals the deep maximum of *R* at the daytime migration depths of mesozooplankton at 90-120 m as well as the persistent maximum of the diapause layer within the deeper layer at 125-140 m. Notably, the depths of the persistent SSL change by approximately 5 m from night to the daytime, while they completely overlap when considered versus the density. Such variations in the depth of the SSL might be linked to inertial oscillations (Ostrovskii et al., 2018).

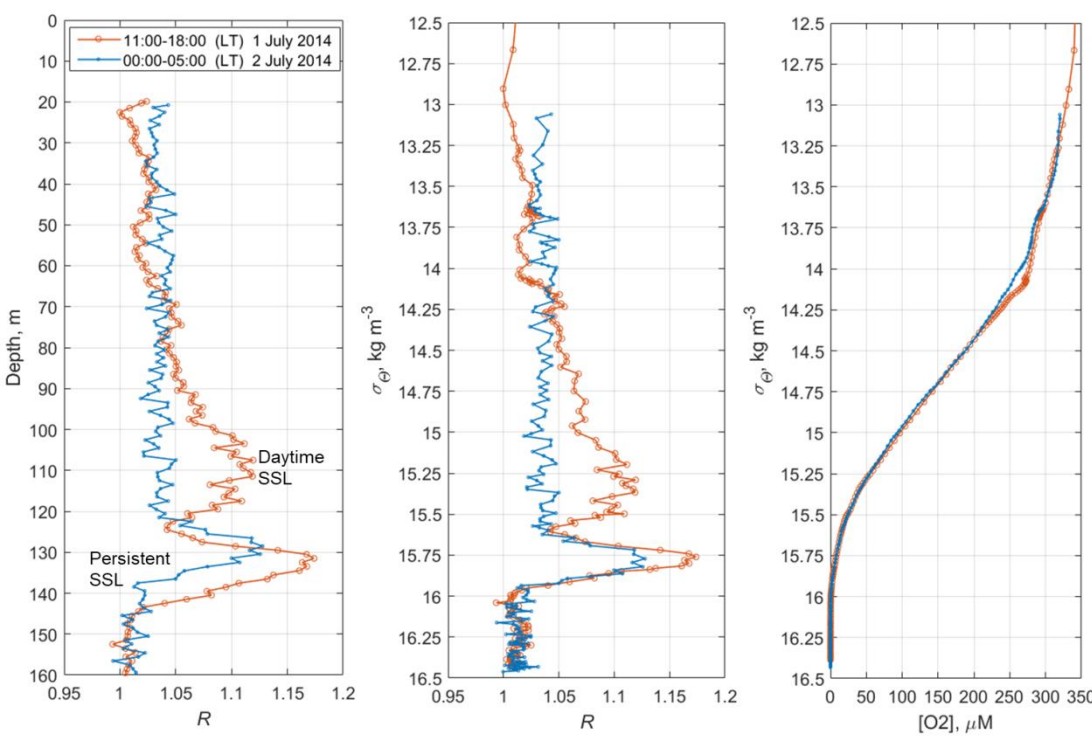

369

**Figure 11: The time averages of R and [O2] for the daytime of 1 July2014 (red), and the nighttime of 2 July 2014 (blue), when net sampling (Fig. 10) took place. Left - The depth profiles of the time averages *R*. Middle – the daytime and nighttime averages *R* versus the specific density, $\sigma_\Theta$. Right – Distribution of the daytime and nighttime averages of the dissolved oxygen concentration versus $\sigma_\Theta$.**

### 3.3 The seasonal variation in mesozooplankton dynamics in relation to dissolved oxygen concentration

In subsection 3.2, it was shown that the mesozooplankton species float on isopycnals in the lower part of the oxycline and in the hypoxic zone. Both the diapausing aggregations and the daytime aggregations are displaced coherently by near-inertial waves. The deep aggregations of mesozooplankton are bounded by certain isopycnal surfaces and iso-oxylines.

Since the oxygen stratification strongly depends on the density stratification in the pycnocline (e.g., Vinogradov and Nalbandov, 1990, Codispoti, et al., 1991, Konovalov et al., 2005, see also example at Fig. 12), it becomes possible to switch from the depth profiles of the directional acoustic backscatter ratio $R(z)$, where $z$ is the depth, to the $R([O2])$ profiles to investigate the seasonal changes of the sound-scattering mesozooplankton layers in terms of *R* versus [O2].

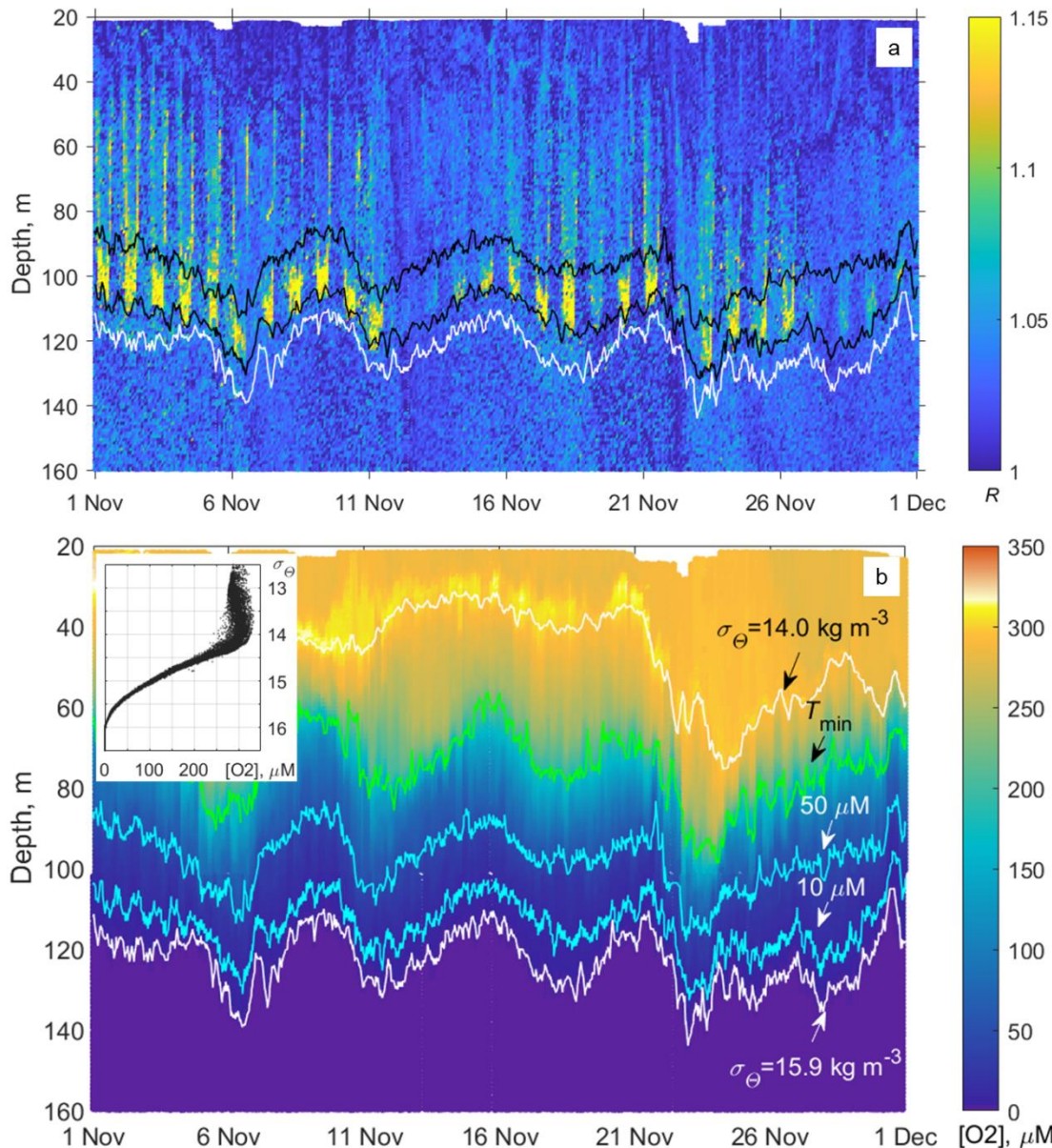

383

**Figure 12: a – Example of the monthly long time series of the vertical profiles of the directional acoustic backscatter ratio $R$ for November 2019 (in total 960 profiles from the moored profiler survey). The upper and lower black lines are iso-oxylines of 50 and 10 μM, respectively. The white line indicates isopycnal $\sigma_\Theta$ = 15.9. b – Evolution of the dissolved oxygen at the profiler mooring site in November 2019. The colored lines indicate the following: the depths of the isopycnals $\sigma_\Theta$ = 14 and 15.9 (top and bottom white lines); the depth of the temperature minimum (green line); and [O2] = 50 and 10 μM (blue lines). The inset shows the diagram of the concentration of dissolved oxygen versus the potential density, [O2]-$\sigma_\Theta$, plotted from the moored profiler data of November 2019. In this example, as well as for other observational periods, the concentration of dissolved oxygen deviates very little from isopycnal surfaces in the lower part of the oxycline where [O2] < 200 μM.**

The average monthly profiles of $R([O2])$ were constructed from $R(z)$ and $[O2](z)$ data for every month when the data
were available. To compute the averages, the daytime was defined as a period beginning 2 h after the local time of sunrise (at
a given date) and ending 2 h before sunset. The nighttime was defined as a period beginning 1 h after sunset and ending 1 h
before sunrise. Example plots of the average profiles $\langle R([O2])\rangle$ computed as arithmetic and bootstrap mean values along
with 95% bootstrap confidence intervals are shown for November 2019 in Fig. 13. In the hypoxic zone, the average values
$\langle R([O2])\rangle$ for the daytime are significantly higher than those for the nighttime. The daytime averages $\langle R([O2])\rangle >$
1.06 were in the range of $[O2] = 9\text{-}40$ µM in November 2019.

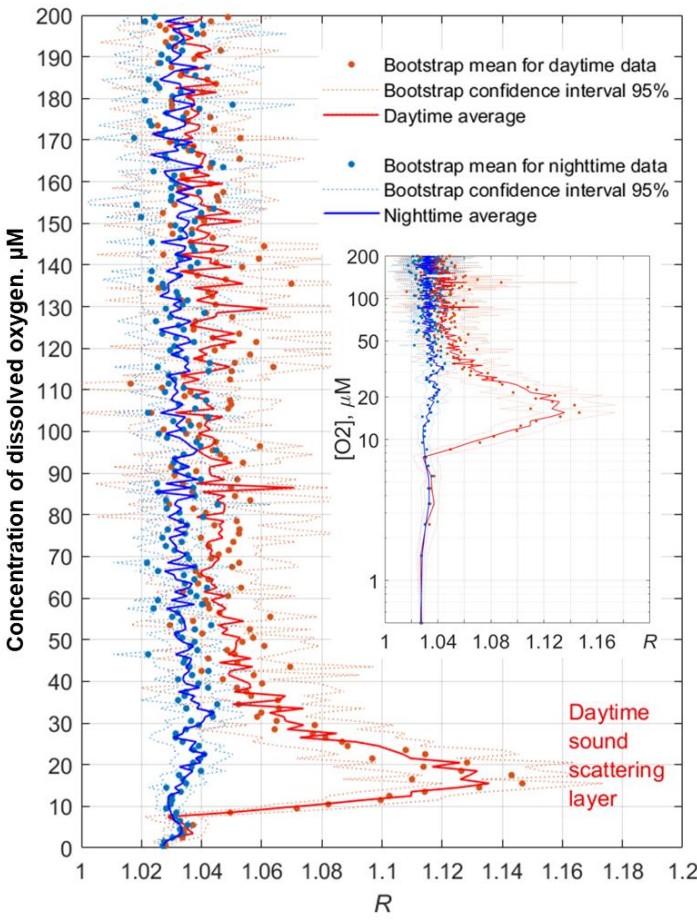

**Figure 13: Example profiles of the daytime and nighttime averages $\langle R([O2])\rangle$ in November 2019. The inset shows the same plots**
**with the Y-axis drawn logarithmically to reflect the lower parts of the profiles (the hypoxic zone) in more detail.**
The average monthly $\langle R([O2])\rangle$ profiles show the seasonal evolution of the mesozooplankton distribution (Fig. 14).
The SSLs are barely discernible in January. One can note some activity in the upper part of the oxycline in February.
Although we unfortunately do not have data for March, in April, two peaks appear in the $\langle R([O2])\rangle$ profiles in the layers
where the concentration of dissolved oxygen is 25-60 µM and 4-9 µM. These maxima correspond to the daytime
mesozooplankton aggregations and the diapause layer, respectively. The upper maximum of $\langle R([O2])\rangle$, which corresponds
to the daytime aggregations of mesozooplankton, may weaken in June-July. However, it becomes stronger again at the end
of summer and in autumn. The largest value for this maximum over the entire observation period $\langle R([O2])\rangle = 1.18$ is
observed in October. At that time, the maximum shifts into the layer where [O2] is 10-25 μM. In December 2019, this peak
was between the 10 μM and 30 μM iso-oxylines.
The maximum of diapause mesozooplankton was strongest in May and July 2020, reaching almost 1.2 at [O2] = 5-8
μM. In August, the diapause mesozooplankton layer shifts in the lower part of the suboxic zone where [O2] = 3-7 μM. It
becomes substantially weaker in September. In October, this layer degrades further. In November, it tends to disappear.

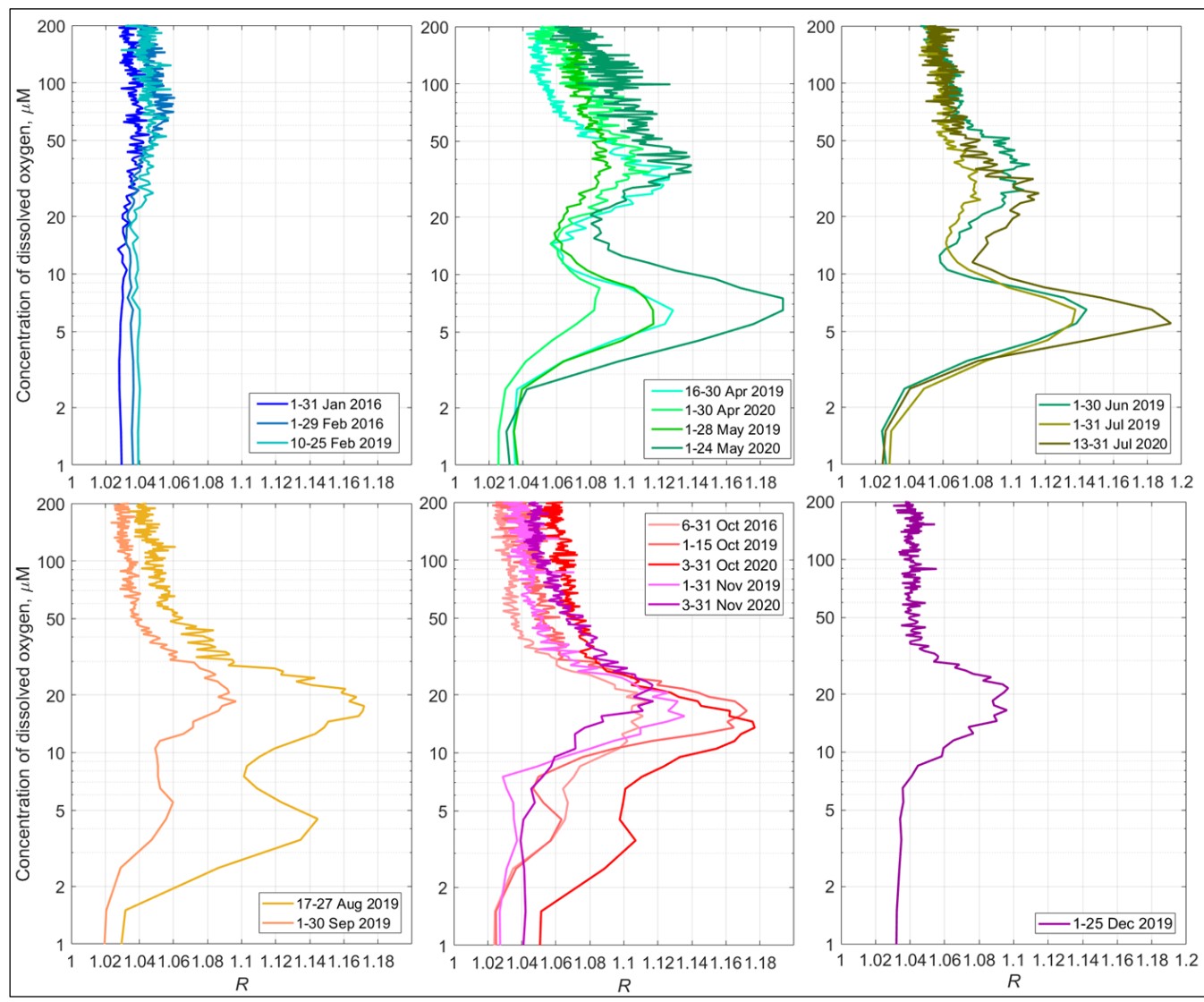


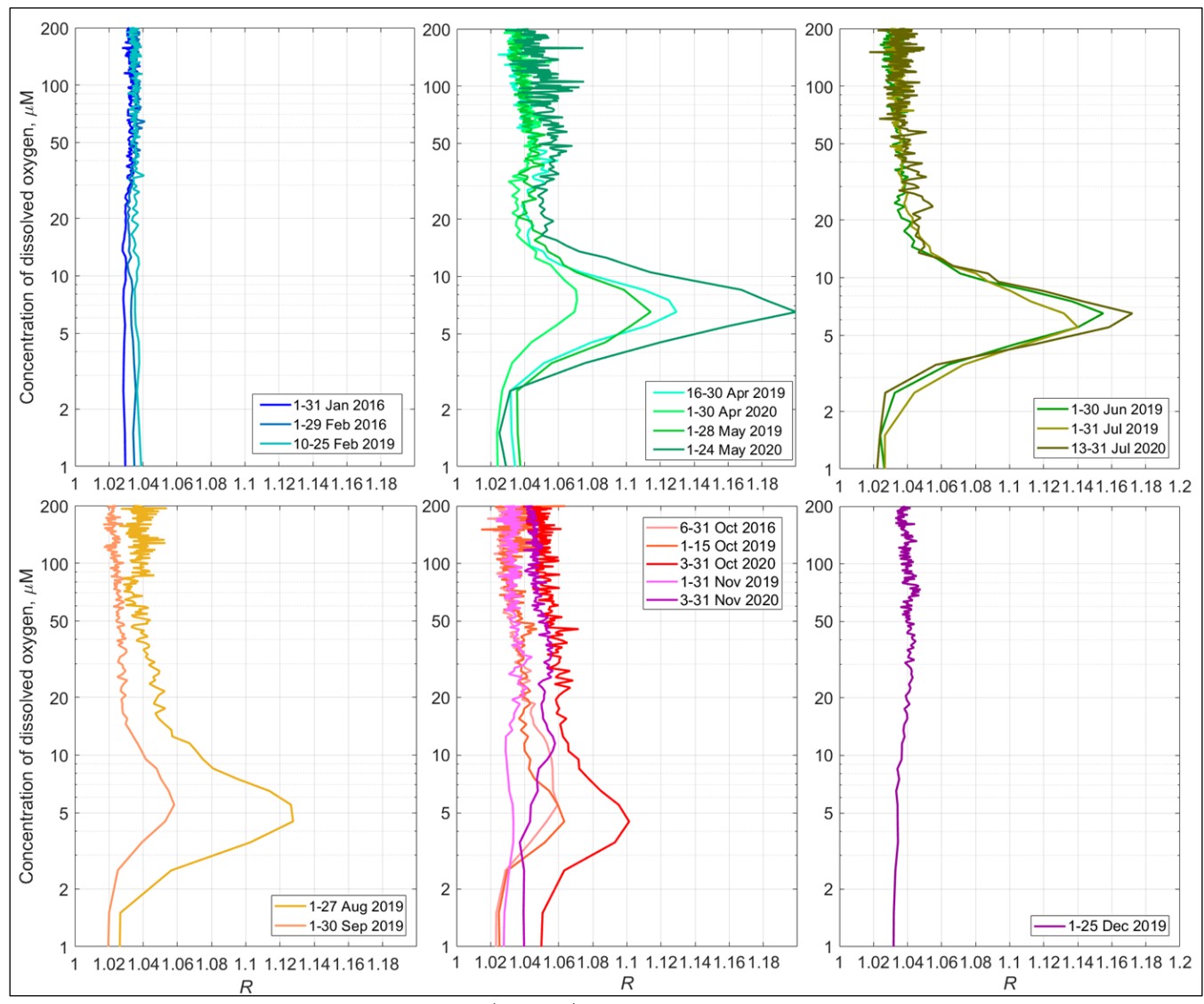

**Figure 14: Top - The monthly averaged profiles of $\langle R([O2])\rangle$ for the daytime over the upper part of the continental slope near**
**Gelendzhik in the NE Black Sea. Bottom – The same for the nighttime.**

## 4 Discussion

### 4.1 Visualization of the sound-scattering mesozooplankton aggregations

Previously, acoustic measurements at a frequency of 2 MHz were not considered a tool for observations of the
mesozooplankton SSLs in the sea due to the limited range of soundings. However, with the advent of ocean profilers with
acoustic Doppler current meters, such as the Nortek Aquadopp, it has become possible to obtain the depth profiles of the
volume scattering strength at 2 MHz frequency in the entire water column and to study the vertical distribution of
zooplankton, such as those in the Black Sea (Ostrovskii and Zatsepin, 2011; Pezacki et al., 2017). Acoustic sounding of
mesozooplankton at two angles is made possible by using the side-looking head of the Nortek Aquadopp instrument. The
combination of horizontal and tilted beam signals allows, on the one hand, eliminating the patches of particles and equalizing
the background scattering level of the echogram and, on the other hand, determining the preferred orientation of
mesozooplankton species migrating through the oxycline. Earlier, Stanton and Chu (2000) reproduced the influence of the
orientation of a 3-mm calanoid copepod (modeled as a high-resolution approximation of an animal profile) on the acoustic
target strength at 2 MHz with respect to an incident sonar beam. The reduction was found to be 5-15% when copepod
orientation was shifted from 0° (broadside incidence) to 30-60°. Benfield et al. (2000) carried out field observations using
the Video Plankton Recorder on George Bank and showed that most *Calanus finmarchicus* (75%) in the depth range of 10-
70 m were within ±30° of the prosome-up or prosome-down orientation. It was suggested that one reason for the behavior
underlying the head-up orientation pattern might be due to the predator avoidance strategy aimed at reducing the
conspicuousness of *C. finmarchicus* when viewed from above. Such individuals would present a significantly reduced cross-
sectional area to an echo-sounder's transducer with correspondingly diminished target strength. It was concluded that it is
necessary to know how the orientation of individuals changes with depth to correctly account for the biomass of
mesozooplankton. Experiments using a multiple-angle acoustic receiver array on live copepods and mysids in a laboratory
tank showed that it is possible to use the scattered acoustic signal to distinguish among zooplankton taxa (Roberts and Jaffe,
2008). Reflections in the frequency range from 1.5 to 2.5 MHz were recorded from untethered 1 – 4 mm calanoid copepods
and 8 – 12 mm mysids over an angular range of 0–47°. That study demonstrated the utility of a multiple-angle acoustic array
for zooplankton identification.
To distinguish the SSLs against the background patterns of vertical flow of settling particles and to study the
orientation of zooplankton species, we propose a simple method for the processing of ultrasound sensing data at three angles.
This acoustic 3-beam geometry provides a partial pragmatic solution for the quest towards the multiple-angle scatter
measurements suggested by models (Stanton and Chu, 2000; Roberts and Jaffe, 2007) and laboratory experiments (Roberts
and Jaffe, 2008). Since the late 1990s, researchers' efforts have been focused on creating multichannel instruments to
measure acoustic backscatter (volume scattering strength) at several frequencies, which contain information about the size
composition of the scatterers, since different frequencies bounce off objects of different sizes (Wiebe et al., 2002, Smeti et
al., 2015). Multichannel instruments in conjunction with video cameras are fairly expensive systems that are used for the
identification of mesozooplankton in its natural habitat. Plausibly, a multichannel 3-angle system featuring several relatively
cheap short-range 3-beam acoustic units each operating at an individual frequency when installed on a vertically profiling
carrier would be a very effective tool for visualizing of zooplankton aggregations.
**4.2 The SSLs validated from the stratified net sampling**
Comparison of the Nortek Aquadopp acoustic backscatter observations with the data obtained by stratified zooplankton
sampling showed good agreement of the features of the diel vertical distribution of zooplankton. This was made possible by

sampling narrow depth strata (10-15 m layers) targeting deep-water aggregations visualized on the echograms. The two-layered structure of the aggregations seen on echograms and *R*-graphs in the daytime (Figs. 3, 7, 9) reflected the species composition of zooplankton in these layers (Figs. 4, 8, 10). The deepest layer bounded by isopycnals 15.9 and 15.7 was visible in the suboxic zone all day and night and was formed by diapausing CV *Calanus euxinus*. To some extent, this monospecific layer was contaminated by crustacean exuviae and carcasses, spent females, and zooplankters' remains sinking from the upper layers and apparently retained on the density gradient. The existence of a nonmigrating diapausing stock located in the suboxic layer from mid-spring to mid-autumn is confirmed by observations from submersible Argus (Vinogradov et al. 1985; Flint 1989), by high vertical resolution sampling with 150 l water bottles (Vinogradov et al., 1992), and by zooplankton net sampling (Arashkevich et al., 1998; Besiktepe, 2001; Svetlichny et al., 2009). However, for some unknown reasons, this nonmigrating layer was not detected by ship-borne echo sounders at frequencies of 38 - 200 kHz (Erkan and Gücü, 1998; Mutlu, 2003, 2007; Stefanova and Marinova, 2015, Sakınan and Gücü, 2016), unlike our data obtained by Aquadopp at a frequency of 2 MHz.

The inclusion of diapause (or dormant stage) in the life cycle of all Calanidae species living in high-latitude and temperate environments is well known (e.g., see review Baumgartner and Tarrant, 2017). Having accumulated a large amount of lipids, diapausing copepods descend into deeper ocean layers where they can exist for several months at the expense of energy reserves. Decreased metabolic rate and developmental delay are characteristic features of diapausing copepods. In the Black Sea, a decrease in the metabolic rate in diapausing *C. euxinus* is caused not only by internal physiological reasons but also by hypoxia in their dormant layer. The oxygen consumption rate in diapausing CV *C. euxinus* in hypoxia decreases by almost an order of magnitude, and the rate of ammonia excretion decreases six times compared with those in their active counterparts in normoxia (Svetlichny et al., 1998).

During the daytime, the upper SSL mostly located above $\sigma_\Theta = 15.7$ consisted of four species, copepods *C. euxinus* and *Pseudocalanus elongatus*, chaetognaths *Parasagitta setosa*, and ctenophores *Pleurobrachia pileus*; the latter had a negligible contribution to dry biomass. This assembly had a wide range of body lengths from approximately 1 mm in *P. elongatus* to 22 mm in *P. setosa*. The different species had different swimming speeds. It was also possible that these species had different physiological tolerances to oxygen deficiency. This confirms earlier observations from the manned submersible, which showed that the daytime aggregation of migrating zooplankton had a layered structure: the lower layer was formed by chaetognaths, whereas the older stages of *C. euxinus* were located above, and ctenophores inhabited the upper part of the aggregation (Vinogradov et al., 1985; Flint, 1989). Furthermore, the different developmental stages of copepods *C. euxinus* and *P. elongatus* occupied different depths, deepening as their size increased (Morozov et al., 2019).

In the evening approximately two hours before sunset, zooplankters begin to ascend to the upper layers, where they spend all the dark hours concentrating in the thermocline layer and below it. In this layer, while feeding, they move in different directions and are oriented randomly (Kiørboe et al., 2009), so they cannot be discernible in the *R*-graphs. According to our data, cold-water herbivorous *C. euxinus* and *P. elongatus* only occasionally ascend into the warm UML, mainly inhabiting colder layers rich in phytoplankton (see also the supplement to the paper by Morozov et al., 2019).

Predator chaetognaths *P. setosa* move upward following copepods, their main prey (Drits and Utkina, 1988). The time of zooplankton migration clearly visible on the echograms is confirmed by the results of net sampling and is consistent with other published data (see for references Morozov et al., 2019).

The vertical migration of zooplankton can increase the vertical flow of carbon and thus contribute to the functioning of the biological pump in the ocean (Tutasi et al., 2020). The mesozooplankton that feed at the surface but metabolize and excrete at depth contribute to the transport of organic matter; more quantitatively, this contribution is estimated to be between approximately 10–50% of the local sinking flux of organic particles (Bianchi et al. (2013) and citation therein).

In the lower part of the oxic zone, the vertical displacements of SSLs coincide with the oscillations of isopycnal surfaces (Figs. 5 and 10). The dissolved oxygen concentration profile tightly hinges on the density stratification in the Black Sea since both are basically due to vertical mixing processes (e.g., Ostrovskii et al., 2018). Hence, displacements of the SSLs with regard to the oxy-isolines are much smaller than those versus the depths. The vertical oscillations with a period of approximately 17 h near the mooring site are due to near-inertial waves (Ostrovskii e al., 2018). Irregular changes in isopycnal depths occur due to hydrodynamic events, such as individual internal waves, oceanic fronts, and jets. Occasionally, the isopycnal depth may change by 30-40 m within a day (Ostrovskii and Zastsepin, 2016).

It is unlikely that copepods maintain their positions on certain isopycnal surfaces by swimming, as displacements of such large amplitudes as tens of meters require an additional depletion of energy reserves. A more beneficial strategy would be to adjust their buoyancy to neutral. Having neutral buoyancy in the hypoxic zone, the copepods would not need to spend much additional energy floating up and down following crests and troughs of internal waves while avoiding entrainment into the suboxic layer. Indeed, direct observations from manned submersibles revealed a quiescent behavior of diapausing copepods and their slow response to light and noise produced by underwater vehicles, both in the Santa Barbara basin (Alldredge et al., 1984) and in the Black Sea (M.V. Flint, personal communication). Neutral buoyancy has been hypothesized to be regulated by changes in lipid composition (Visser and Jónasdóttir, 1999); however, Campbell and Dower (2003) argued that this buoyancy regulation mechanism is inherently unstable because wax esters are more compressible than seawater. An alternative mechanism for buoyancy regulation in diapausing copepods that involves the replacement of heavy ions with lighter ammonium ions in hemolymph has been proposed by Sartoris et al. (2010) by analogy with other invertebrates. Later, Schründer et al. (2013) found high concentrations of ammonium ions in the hemolymph of a diapausing species, *Calanoides acutus*, and suggested that these copepods could achieve neutral buoyancy through their biochemical body composition without swimming movements. This mechanism obviously would better explain the observed phenomenon of diapausing copepod movement synchronized with the displacements of the isopycnal surfaces in the Black Sea.

**4.3 Seasonal variations of the deep mesozooplankton SSLs**

For most of the year, from April to October, the sound scattering profiles in the deeper part of the oxic zone were bimodal during the day (Fig. 14), reflecting the vertical distributions of two different zooplankton cohorts, migrating and diapausing.

The oxygen concentration at which the maximum backscattering signal from migrating zooplankton was observed decreased
throughout the year, from ca. 70-90 µM in January-February to 30-40 µM in April-August and to 15-20 µM in September-
December. Two explanations can be considered for the seasonal shift in the preferred oxygen concentration in these
zooplankters. On the one hand, this shift can be attributed to the deepening of the suboxic layer in January-February,
followed by gradual shallowing from April onwards (Fig. 15). If we assume that the depth of daytime zooplankton
aggregation depends to a large extent on the species-specific migration amplitude then with a deepened suboxic layer ([O2]
< 10 µM), the migrating zooplankton will reach shallower depths, and its aggregation will be at a higher oxygen
concentration. Conversely, with a shoaling suboxic layer, zooplankton localize at lower oxygen levels.
This assumption is consistent with the data of Vinogradov et al. (1992), who found the daytime aggregation
maximum of migrating CV and female *C. euxinus* at an oxygen concentration of 18 µM when the suboxic layer was at a
depth of 110 m and at [O2] = 36 µM when the suboxic layer was at a depth below 170 m. This suggests that the trade-off
between the additional metabolic cost for extended swimming and metabolism reduction caused by low oxygen is in favor of
a decrease in the diel migration amplitude of *C. euxinus*. This differs from observations (Wishner et al., 2020) on the
migration of *Lucicutia hulsemannae* in the eastern tropical North Pacific, where this species changes its daytime location in
response to changes in the depth of the oxygen minimum zones (OMZ). For example, at the lower oxycline, the depth of
maximum abundance for *L. hulsemannae* shifted from ∼600 to ∼800 m in an expanded OMZ compared to a thinner OMZ
but remained at similar low oxygen levels in both situations. *L. hulsemannae* is an example of a "hypoxiphilic" species
(Wishner et al., 2020). However, unlike *Calanus* spp., *L. hulsemannae* is a strong swimmer capable of diel vertical migration
with amplitudes as large as approximately 1000 m.
Another explanation for the seasonal shift in the depth of the daytime aggregation in the Black Sea is the change in
taxonomic and age composition of the migrating cohort. A decreasing/increasing share of strong/weak swimmers with
different tolerances to oxygen deficiency may lead to a shift in the depth of daytime aggregation. The oxygen concentration
was in the range of 15-60 µM in the layer of daytime aggregation of migrating species in the Black Sea. Similarly, the
vertical distribution of migrating CV and adult *Calanus chilensis* off northern Peru was characterized by high abundance in
hypoxic waters at oxygen concentrations between 5 and 50 µM (Hirche et al., 2014).



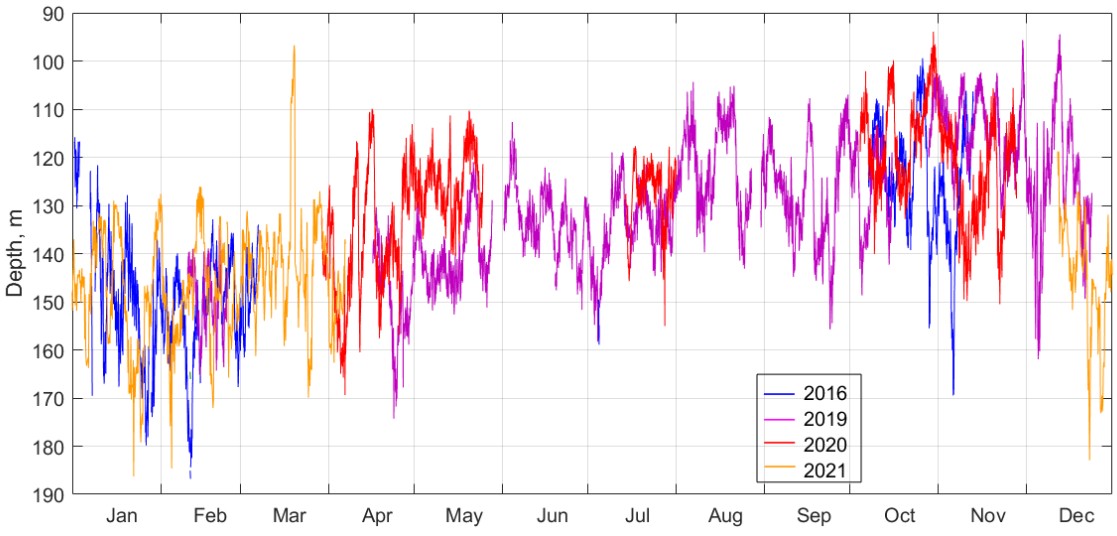


**Figure 15: The suboxic boundary depth ([O2] = 10 μM) as observed by the moored profiler in 2016, 2019, 2020, and 2021.**

554         Based on our data, a diapausing cohort of *C. euxinus* appeared in April and terminated in November (Fig. 14).

According to Vinogradov et al. (1985) and Svetlichny et al. (2009), the diapausing stage of *C. euxinus* was not found in the
Black Sea in March. Hence, it is assumed that the diapausing stock is formed in April, when the offspring of the first
generation of *C. euxinus* develop into the CV and accumulate sufficient lipid reserves. The energy reserve and a decrease in
metabolic rate allow diapausing CV to exist without food for seven months in the suboxic zone (Vinogradov et al., 1992),
which agrees with our observation of the diapause duration. The suboxic zone provides a refuge from large visual predators,
which generally need higher oxygen concentrations (e.g., Bianchi et al., 2013).

561         The diapausing layer is bound by 3 μM and 10 μM and peaks at 5-7 μM oxygen. Earlier, the oxygen survival

threshold for diapausing *C. euxinus* was determined experimentally at [O2] = 1.8 μM (Vinogradov et al., 1992).
Physiological tolerance for hypoxia in diapausing *C. euxinus* resembles that reported for diapausing stage CV *Calanus*
*pacificus,* which formed narrow dense aggregations at an oxygen concentration of 6.25 μM in the Santa Barbara Basin
(Alldredge et al., 1984). The diapausing *C. pacificus* at a depth of 450 m was characterized by quiescent behavior, low
laminarinase activity (as a proxy of feeding), and high lipid storage.

567         Similar tolerance for hypoxia was reported for diapausing *Eucalanus inermis*. Based on lactate dehydrogenase

activity in the deep-dwelling CV and female *E inermis*, their oxygen tolerance threshold was defined at the level of 4.47 μM
in the Pacific Ocean (Flint et al., 1991). Wishner et al. (2020) found a monospecific aggregation of *E. inermis* diapausing at
extremely low oxygen, 1.0–5.7 μM, in the eastern tropical North Pacific. In the Black Sea, the diapause depth is directly
associated with certain density surfaces and consequently with a specific concentration of oxygen. From April to November,
the isopycnal surfaces bounded by the suboxic layer move upward from depths of 130-160 m to 110-140 m. On this seasonal
trend, the superimposed surfaces exhibit strong variations up to 60 m in amplitude at time scales from 17 h to several days

(Fig. 15). Diapause layers varied in depth along with isopycnal oscillations, allowing the copepods to remain in a constant-low-oxygen habitat.

## 5 Conclusions

The key to using high-frequency sound in this study is to deploy the acoustic transducer in a manner that gets it sufficiently close to the animal aggregations of interest. To visualize the mesozooplankton SSLs over the echogram with background vertical flows of settling particles, we take advantage of the differences in acoustic scattering that is isotropic on the settling particles and anisotropic on zooplankton species due to the elongated shape of the animals because their side view area is larger than the head-view area or the tail-view area. The calculations of the ratio $R$ of the volume scattering strength of the horizontal acoustic beams to the volume scattering strength of the slanted beam allow visualizations of the mesozooplankton aggregations of the specimen to be oriented vertically. This three-beam approach enhances the capability of underwater ultrasound sensing to observe the mesozooplankton layers.

Linking the values of $R$ to oxygen concentration enables us to derive the monthly averages from many profiles despite the fluctuations in vertical distribution. The analysis of the oxygen-deficient zone allows us to describe the seasonal evolution of diel zooplankton migrations, to determine the preferred oxygen regime for migrating and nonmigrating zooplankters and to define the timing of formation, termination, and duration of diapause in CV *Calanus euxinus* in the Black Sea.

Aggregations of vertically migrating zooplankton, consisting mainly of the older copepodite stages of *C. euxinus* and *Pseudocalanus elongatus* and large-sized chaetognaths *Parasagitta setosa,* are observed within the hypoxic zone during the daytime and mostly in the thermocline layer at night. The volume scattering strength in migrating SSL in the hypoxic layer varies seasonally, with a minimum in winter and a maximum in late summer - early autumn. The location of this SSL also changes in relation to the oxygen concentration in the range [O2] between 10 and 100 μM. Roughly, the deeper the suboxic zone is located, the higher the oxygen concentration at the layers where the migrating species are aggregated. These variations are hypothesized to address seasonal changes in the taxonomic and age composition of migratory zooplankton. The maximum depth of zooplankton vertical diel migration is limited by the upper boundary of the suboxic zone ([O2] = 10 μM.

The nonmigrating diapause SSL is observed at low [O2] = 3-10 μM from the beginning of April to the end of October, suggesting a seven-month duration of diapause in *C. euxinus*. This persistent layer does not exceed 5-10 m in thickness. The volume scattering strength in this monospecific layer may exceed that in the overlying daytime SSL, apparently indicating the tighter aggregation of diapausing copepods compared to the aggregations of multispecies migrating zooplankton. Diapause layers vary in depth along with isopycnal oscillations, allowing copepods to remain in a constant-low-oxygen habitat.

Fluctuations of the SSLs are subject to interannual changes. It is necessary to maintain moored profiling acoustic
observatories in the Black Sea for a detailed analysis of year-to-year variability.

**Appendix**
**Is the mesozooplankton specimens' vertical orientation tilted in the deep aggregations?**
In the deep aggregations, the mesozooplankton species, while being oriented vertically in general, might be tilted with
respect to the vertical axis. Furthermore the specimens are probably occasionally tilted i.e., their azimuth angles are
distrusted randomly so that the broadside incident angles of the horizontal acoustic beams would dominate the acoustic
backscattering data. Since the horizontal beams of the Nortek Aquadopp instrument are orthogonal, one can calculate the
standard deviation of the ensemble of the ratio of the acoustic backscatter of horizontal beams $A_1/A_2$ to check for the
possibility of a tilt. There is a high probability that $\langle A_1/A_2 \rangle = 1$ for the aggregations of tilted species, while the standard
deviation should be greater than 0. The persistent SSL caused by the diapausing mesozooplankton appears on the spring and
summer profiles of $R([O2])$ as the maximum in the layer where $[O2] < 10$ μM, i.e., in the suboxic layer. The orientation of
the mesozooplankton species in this layer is mostly vertical but tends to be slightly more tilted than in the daytime
aggregations of migrating mesozooplankton, as indicated by the $A_1/A_2$ ratio (see example for July 2019 at Fig. A1).

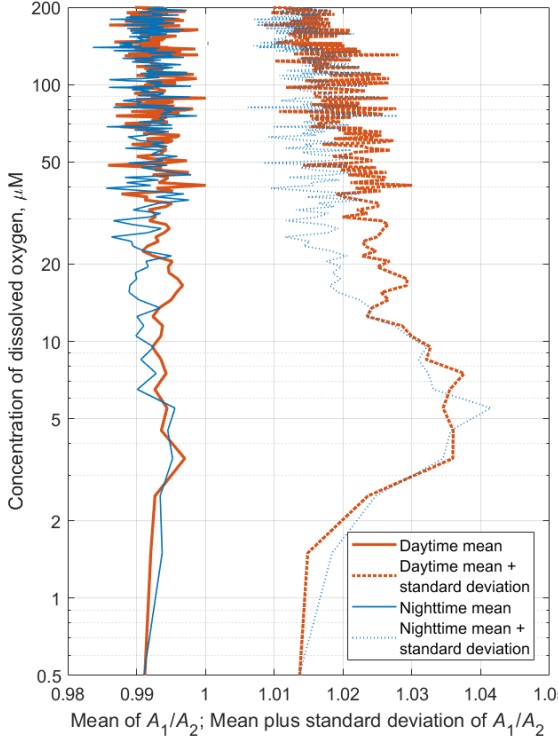


**Figure A1:** Monthly averages of the depth profiles of the acoustic backscattering amplitude ratio $A_1/A_2$ for the time series of the daytime (solid red line) and nighttime (blue solid line) data in July 2019. Dotted lines indicate the values of the standard deviations from the means.

*Data availability.* Underlying research data can be accessed via HTTPS://DOI.ORG/10.13140/RG.2.2.28470.73285, License CC BY-NC 4.0 and HTTPS://DOI.ORG/10.13140/RG.2.2.27548.62084, License CC BY 4.0.

*Author contributions.* AO analyzed the moored profiler Aqualog data and wrote the parts of the paper related to the profiler mooring measurements and data analysis. EA analyzed the net zooplankton data and wrote the parts of the paper related to the zooplankton distribution analysis. VS deployed the mooring and handled the profiler sensors. DS is a design engineer of the profiler who also maintained the profiler.

*Competing interests.* The authors declare that they have no conflict of interest

*Acknowledgements*. We thank A. Zatsepin for promoting of the moored profiler measurement program in the Black Sea. The Black Sea field study is carried out in the framework of Russian Ministry of Science and High Education Assignment No. 0128-2021-0016. The net sampling data analysis is supported by Russian Science Foundation grant No.20-17-00167. The Aqualog profiler data processing and analysis are supported via grants No. 19-05-00459 and 19-45-230012 by Russian Fund for Basic Research and Krasnodarsky Kray Ministry of Science, Education, and Youth Policy. We are grateful to editor and three anonymous referees for helpful comments on this paper.

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
