# Peer review of "Seasonal variation of the sound-scattering zooplankton vertical distribution in the oxygen-deficient waters of the NE Black Sea"

_Ocean Science, 2020_

## Referee Comment (RC1) · Anonymous Referee #1 · 26 Dec 2020

The manuscript (MS) showed a pure acoustical scattering layer daily migrating using an autonomous water current profiler operating at 2 MHz during months of some years in the Black Sea. The Black is very good environment to study the zooplankton acoustics because of less diversified organisms and daily layered migration of each species in separate. There are some studies published already on this topic as inferred from the introduction of the MS. The MS however has not covered the scientific aspects rather different than the results of the previously published papers using the echosounder and ADCP because of the main problems presented in the MS as follows: Introduction of the MS did not present lack of the other studies and their innovations to purpose the annual cycle of the SL with the reasons. Material Methods would have data analyses

and processing methodology with the methodical terms and study area description for the oxygen and other physical parameters which confine the DVM, e.g. the R and its distinguished importance from the acoustical energy , relationships between the orientation and each transducer of the profiler, acoustical intensity or amplitude calculation, removal of unwanted targets else (fishes, particles, marine snows and some untargeted individuals nearby targeted species relative to frequency) than the zooplankton, detection range of the frequency and dynamic ranges proportion to the aforementioned zooplankton. Results showed only pure SL moving up/downward and staying at constant depths during the days of the different months. On surface SL, one or two SL migrating daily, and one DSL staying at constant depth in time looking at the R terms which lack of importance and distinguished description of the other scattering energy units. The 2 MHz was expected to detect particles in size down to 0.2 mm in diameter equivalent to the spherical particles regardless of the beam pattern of the organisms. The beam pattern of the organism is needed to outdraw depending on the orientation and shape during the both direction migration and duration at deep depth and surface, shall the three transducers particularly A3, help information on the body shape and orientation. Some questions are arisen to be explained so; The 2 MHz must detect many species but only one or sometimes two SL were observed during the diel movement. As a consequence, some studies showed two scattering layers belonging to two different species of fluid-like organisms depending on their acoustical reflection coefficients in the Black Sea (copepods, Cheatognatha) as well as significant detection of moon jellyfish using rather lower frequencies than the frequency used in the MS. Main of the results and findings have showed one scattering layer migrating during the day, which could be more number of scarring layer in the present study. The SL was compared only with dissolved oxygen and not other physical parameters such as sigma-t of water density which describe the DVM in the Black Sea, and study area was well described for the regional differences such as upwelling or downwelling, rim currents which course the DVM and their speed across oxygen and water density so depth of water column. Daily differences could be ignored using the water density along the

rim currents. Some vertical lineations occurred from surface down to greater depth of the SL which could be attributed presumably to the particle sedimentations. The SL amplitude as counts showed daily differences in same months without a reasonable explanation. Mostly, one SL migrated between surface and a certain depth above the minimum oxygenated layer, not reaching the hypoxia layer, but other studies observed different DVM reaching deep layers as well as staying at minimum oxygen layer during diapausing and daytime. The R values are expected to changes in time during the DVM, because mainly of the changes in individual swimming speeds and organisms concentration insonified volume by the acoustics. Therefore, what is importance of the R to denote significance in the intensity of the SL. The swimming speed through the oxygen concentration is one of the recognition parameters identical for some species migrating during the day in the Black Sea. When the DSL arrived non-migrating SL at sub-surface, there was however no aggregation of the SL during the night. Inherently, some scattering layers must occur at sub-surface even if the orientation of the organisms change over there. What is the difference between Fig. 11 and 12 as well as Fig. 9, one was average of amplitude of A1 and A2, the next one is based on their ratio through the same water column denoted with the oxygen concentrations. Indeed, both figures contain zooplankton migrating upward and downward through the same water column in order to show the orientation of the organisms. They had similar information. Breaking apart from the DVM SL, some SL returned back or stopped going down at the middle way of the water column during the DVM as shown in Fig. 10. In general, the echogram data show revealing much information remained unexplained because of lacked one of the ground-truthing methods, discrete layered zooplankton samples. Discussions were not well written to justify the findings, observations and acoustical parameters used in the MS. Most of the results were postulated to the assumptions for the justification. One DVM SL was predominated in the present study, seemed to be typical characters of Calanus euxinus' DVM from a region of a downwelling zone, not reaching the minimum oxygen layer of the Black Sea. Juday net samples are missing to show the DVM of the zooplankton Calanus euxinus and Pseudocalanus elongates,

which are claimed to be observed by the acoustics, but one DVM SLs are presents overall in the MS. The swimming speed estimated in the MS was discussed with those of the other studies, but there were no data for the swimming speed of the zooplankton in the results MS. Why other zooplankton which contain the similar body material properties to two targeted copepod species did not appear in the SL of the DVM using the very high frequency, even though the results were discussed for the two mesozooplankton. Such questions could be clarified already in the MS but the English of the MS is not comprehensible to me.

---

## Referee Comment (RC2) · Anonymous Referee #2 · 15 Jan 2021

Comments to MS os-2020-106. "Annual cycle of sound-scattering mesoplankton in the oxycline and hypoxic zone in the northeastern Black Sea" by Ostrovskii et al.

This manuscript presents an interesting dataset taken in the hypoxic zone of the Black Sea. The authors have undoubtedly done a good amount of data collection, analyses, and present their results and conclusions in a reasonable way. However, in its current form, the manuscript still requires further revision according to comments from the another reviewer (RC1) with which I fully agree.

Some aspects that I consider authors should improve, clarify and justify in a better way are:

[Figure]

1) The title is on the annual cycle, but nowhere the annual cycle was actually shown. So, what do authors understand by annual cycle? Authors have taken data from 2013 to 2020 which sounds great, but they have not estimated the annual cycle. I advise authors to perform these calculations in order to affirm the title of their manuscript or simply delete and rewrite it again.

2) The abstract must be completely rewritten since it is not consistent with the classic and logical way of establishing the purpose and motivations of the study, then some background indicating the main results and conclusions.

3) A weak point in the work is the lack of simultaneous sampling of zooplankton. If this was done, authors should show their time series with these samples to validate the acoustic records. If these samples are already in previous papers then authors should emphasize this point throughout the text. This important aspect is not very clear.

4) The Introduction section should be rewritten. The backgrounds, motivation, objectives and hypothesis that were tested are missing. For example, paragraph from line 41 should be earlier in the text.

5) Line 70. Correct to "species".

6) Line 92. Please indicate authors or doi (if applicable) rather than insert manual. The same for lines 98 and 99.

7) Line 120. The authors say "... transducer is most sensitive to particles with a diameter of 0.23 mm..." then, it is necessary to make a simple comparison between this particle diameter with the respective diameter of the copepod species. This would help to clarify ideas and further support its acoustic validation.

8) Line 133. Please avoid unconventional symbols or nomenclature. Simply indicate 1 m instead of "10ˆ0".

9) Line 143. Do you have observational evidence that the same zooplankton aggregations were sampled? If so, authors should show it as results or if it is an assumption to

mention it as such throughout the MS.

10) Line 155. Please indicate accuracy and detection limit for oxygen sensors. This point is important since in several figures there are very low O2 values, so these values are questionable if the detection limit of the instrument is not known. If this is the case, the values below the detection limit must be deleted.

11) Line 179. Please explain in more detail or reference about the hydrogen sulfide zone.

12) Line 189. The "R ratio" should be explained in more detail. Perhaps one way to start is by inserting an equation and defining the parameters one by one. Furthermore, authors should emphasize the rationale for using this ratio when other methods are available in acoustic measurements.

13) Line 195. Correct "daythe"

14) Fig. 4. Why do the authors define the hypoxia zone based on density values and not on O2 concentration values? Not bad, but for comparative and conventional purposes it is more useful to define this zone with O2 values.

15) Line 206. Delete "compare"

16) Lines 219-227. This paragraph should be in the discussion section.

17) Figure 7 refers to October 2004... other figures to November 2019. Why this difference? Authors must give justified reasons to present their results in this way.

18) Figures 8, 9, 11-14. The O2 concentration values are in power format. Please avoid this form and use conventional format. The transformed scale is justified by emphasizing on curves, but numbers not.

19) Figure 9. Is it about the entire time series or some selected months? Please clarify.

20) Figure 10. It refers to June-July 2014, why? All these discrepancies in presentation

of results into figures without adequate justification and methodological clarification make difficult to understand the main message of the MS.

21) Figure 11. Refers to August 2019. . ..

22) Line 288. Without detailed information on the detection limit of O2 sensors, it is difficult to accept the values indicated here, i.e. 4-9 uM.

23) Lines 318-319. Fluorescence was not measured in this study, if it was, it should be indicated accordingly. If there are support from other sources, they should be indicated.

24) Figure 14. This figure is fine as a corollary to the main message. I like it, although in 3D view it is somewhat difficult to visualize. Perhaps authors could rotate angles a bit more for better visualization or separate information into two 2D panels.

25) Lines 443-444. I do not understand what authors are trying to say in this sentence. Please rewrite.

26) Line 449. Researchgate is a popular and excellent platform to disseminate scientific research; however, to deposit datasets I suggest to use platforms specifically designed for this purpose. Please consider it.

27) Finally, consider to reduce the length of the conclusion paragraph.

---

## Referee Comment (RC3) · Anonymous Referee #3 · 3 Feb 2021

The manuscript (MS) presents a modern imaging techniques such as the acoustic of pelagic communities with advantages to be informative about heterogeneity and transcend multiple spatial scales. The article is based on a large data set (2013-2020) obtained from the application of an alternative innovative approach - a moored Aqualog profiler equipped with an ultrasound probe, a conductivity-temperature-depth (CTD) probe, and a fast oxygen sensor with the advantage of frequent year-round measurements of collocated vertical profiles of sound scattering, temperature, salinity, and oxygen concentration in the water column from the near-surface to the bottom layer with a high vertical resolution. This topic is not novel but the previous studies are based on ship-borne echograms. The authors clearly indicate their own original contribution.

The work is interesting, results are sufficient and the paper addresses scientific questions within the scope of OS but needs some revisions. 1) The abstract should be condensed and concentrated around the main aim, results and conclusions. 2) In the introduction the main sound-scattering zones are defined according to Ostrovskii and Zatsepin (2011) but I suggest to bind them with the density sigma theta which is relevant to the mesozooplankton vertical distribution especially for the Black Sea. As a consequence, it needs to be developed and compared in the results and discussion chapters. In the MS the lowest depth mentioned was at $\sigma\ddot{I}t' = 15.9$ kg m-3. However, in other studies (Mutlu 2007a, b,) sigma theta - 16.2 kg.m-3 , identified as oxygen minimum zone (OMZ) (Tugrul et al. 1992), is a layer where Calanus euxinus spend their daytime. How will the authors comment these differences? 3) The authors presented different seasonal variation in mesoplankton dynamics in relation to dissolved oxygen concentrations. Additionally the SL amplitude showed differences in same months but a reasonable explanation is not presented. 4) There are two dominant species well acoustically discriminated in the Black Sea – Calanus euxinus and Parasagitta setosa (Mutlu 2007) but the later was not included in the MS which need an explanation. 5) Line 315 The authors say "…………..two layers in the cold intermediate layer (CIL) (temperature less than 9°C),………….." but according to the literature the positions of the 8°C isotherms have traditionally been considered the lower and upper boundaries of the CIL (Blatov et al., 1984; Ozsoy and Unluata, 1997). Winter cooling, which is an essential element of the seasonal variability could be used for comparison of unlike SL profiles in the same season (month) in different years. 6) Conclusions should be rewritten - shortened, concentrated and clearer, emphasizing the research contribution. 7) Correction: Pseudocalanus elongatus (WoRMS) is the right species name, not Pseudocalanus elongates 8) Figure 3 It is mentioned that "The horizontal axis represents UTC time." Please, check. 9) References should be checked. For example, Arashkevich et al. 2014 (in the text) Arashkevich et al. 2013 (in the reference list); Arashkevich et al. 199, Besiktepe et al., 1998 are missing in the reference list but are cited in the MS and etc. 10) The language should be precise.

---

## Author Comment (AC1) · 10 Mar 2021

Thank you for the review of our manuscript. We fully agree with your comment about the importance of data zooplankton sampling for verification of the acoustic backscatter data. The manuscript was revised to address these kinds of questions. We invited Dr. Elena Arashkevich to share with us the Juday net data obtained nearby the profiler mooring. She also contibuted her results of analysis of the mesozooplankton species in the samples. We also extend the analysis with the new acoustic data collected most recently (summer-autumn 2020). The manuscript is rewritten in line with yours and the other reviewers' comments. Errors are corrected in the manuscript, the figures are

partly replaced, and some materials are brought into the appendix. Below we list your comments together with our responses to them.

Comment #1: Introduction of the MS did not present lack of the other studies and their innovations to purpose the annual cycle of the SL with the reasons.

Response: Section Introduction is rewritten to review more comprehensively the current status of the mesozooplankton research in the Black Sea and to indicate more clearly the goals of our study.

Comment #2: Material Methods would have data analyses and processing methodology with the methodical terms and study area description for the oxygen and other physical parameters which confine the DVM, e.g. the R and its distinguished importance from the acoustical energy, relationships between the orientation and each transducer of the profiler, acoustical intensity or amplitude calculation, removal of unwanted targets else (fishes, particles, marine snows and some untargeted individuals nearby targeted species relative to frequency) than the zooplankton, detection range of the frequency and dynamic ranges proportion to the aforementioned zooplankton.

Response: The simple method for processing the Nortek Aquadopp data of ultrasonic sounding of the water column at three angles is based on earlier model and laboratory studies (Stanton, Chu, 2000; Roberts and Jaffe, 2007, Roberts and Jaffe, 2008). It allows to distinguish the mesozooplankton sound-scattering layers against the background of vertical flows of settling particles while taking advantage of the fact that the acoustic scattering is isotropic on the settling particles and anisotropic on zooplankton species due to the elongated shape of the animals because their side view area is larger than the head-view area or the tail-view area. For the Nortek Aquadopp acoustic Doppler current meter, the data on sound scattering amplitudes is essentially a by-product. The key to using high-frequency sound in this study is to deploy the acoustical transducer in a manner that gets it sufficiently close to the animal aggregations of interest. Notice, the orientation of the animals in the deep aggregations is poorly

explored (except e.g., Dagg, M. J. (Sinking particles as a possible source of nutrition for the large calanoid copepod Neocalanus cristatus in the subarctic Pacific. Deep Sea Research I. 40, 1431–1445, 1993), Kiørboe T. (How zooplankton feed: mechanisms, traits and trade-offs. Biol. Rev, 86, pp. 311–339. 311. doi: 10.1111/j.1469-185X.2010.00148.x, 2011), see also unpublished report by Ashjian et al. https://www.researchgate.net/publication/266329080_Spatial_and_Temporal_Variability_of_Zooplankton_Thin_Layers_Th 2008)). Our study qualitatively indicates that the mesozooplankton species basically maintain vertical orientation in the deep aggregations in the Black Sea. Unfortunately Nortek Co. does not provide any calibration data on the Aquadopp so one cannot assess relevant acoustic characteristics that you mentioned. However, empirically, we realized that the Aquadopp observational data can be useful for research on the mesozooplankton if the data is processed in the way that we suggested.

Comment #3: Results showed only pure SL moving up/downward and staying at constant depths during the days of the different months. On surface SL, one or two SL migrating daily, and one DSL staying at constant depth in time looking at the R terms which lack of importance and distinguished description of the other scattering energy units. . . The beam pattern of the organism is needed to outdraw depending on the orientation and shape during the both direction migration and duration at deep depth and surface, shall the three transducers particularly A3, help information on the body shape and orientation.

Response: The beam pattern is shown on Fig. 1. Since the Aquadopp instrument moves up and down along the mooring line, the beam pattern remains unchanged through the water column. Our interpretation of the empirical data is that the orientation of the mesozooplankton species is changed from the random in the upper part of the oxic zone ( [O2] > 200 mkm) to the vertical in the oxygen-deficient zone.

Comment #4: The 2 MHz was expected to detect particles in size down to 0.2 mm in diameter equivalent to the spherical particles regardless of the beam pattern of the organisms.

Response: The mesozooplankton species composition in the deep aggregations is presented in detail in the revised ms. The typical sizes of the mesozoopoankton species are given at the end of subsection 3.1.

Comment #5: The 2 MHz must detect many species but only one or sometimes two SL were observed during the diel movement. As a consequence, some studies showed two scattering layers belonging to two different species of fluid-like organisms depending on their acoustical reflection coefficients in the Black Sea (copepods, Cheatognatha) as well as significant detection of moon jellyfish using rather lower frequencies than the frequency used in the MS. Main of the results and findings have showed one scattering layer migrating during the day, which could be more number of scarring layer in the present study.

Response: The sound-scattering layers (SSLs) are validated via the net zooplankton sampling in the revised ms.

Comment #6: The SL was compared only with dissolved oxygen and not other physical parameters such as sigma-t of water density which describe the DVM in the Black Sea, and study area was well described for the regional differences such as upwelling or downwelling, rim currents which course the DVM and their speed across oxygen and water density so depth of water column. Daily differences could be ignored using the water density along the rim currents. Some vertical lineations occurred from surface down to greater depth of the SL which could be attributed presumably to the particle sedimentations. The SL amplitude as counts showed daily differences in same months without a reasonable explanation. Mostly, one SL migrated between surface and a certain depth above the minimum oxygenated layer, not reaching the hypoxia layer, but other studies observed different DVM reaching deep layers as well as staying at minimum oxygen layer during diapausing and daytime.

Response: The SSLs are compared with the water density vertical distribution in the revised ms. The main objective of our study is the temporal (seasonal) rather than

spatial variability.

Comment #7: The R values are expected to changes in time during the DVM, because mainly of the changes in individual swimming speeds and organisms concentration insonified volume by the acoustics. Therefore, what is importance of the R to denote significance in the intensity of the SL. The swimming speed through the oxygen concentration is one of the recognition parameters identical for some species migrating during the day in the Black Sea. When the DSL arrived non-migrating SL at sub-surface, there was however no aggregation of the SL during the night. Inherently, some scattering layers must occur at sub-surface even if the orientation of the organisms change over there.

Response: We agree that the estimates of the DVM speed based on the acoustic data does not account for the fact that the different components of zooplankton have different swimming speed also the different species start the DVMs from different depths. So in the revised ms, we decided to remove the relevant part of the discussion.

Comment #8: What is the difference between Fig. 11 and 12 as well as Fig. 9, one was average of amplitude of A1 and A2, the next one is based on their ratio through the same water column denoted with the oxygen concentrations. Indeed, both figures contain zooplankton migrating upward and downward through the same water column in order to show the orientation of the organisms. They had similar information.

Response: Fig. 12 is brought into Appendix and Fig. 9 is deleted.

Comment #9: Breaking apart from the DVM SL, some SL returned back or stopped going down at the middle way of the water column during the DVM as shown in Fig. 10. In general, the echogram data show revealing much information remained unexplained because of lacked one of the ground-truthing methods, discrete layered zooplankton samples.

Response: The net zooplankton sampling data are added extensively in the revised
manuscript to validate the acoustic backscatter observations.

Comment #10: Discussions were not well written to justify the findings, observations and acoustical parameters used in the MS. Most of the results were postulated to the assumptions for the justification. One DVM SL was predominated in the present study, seemed to be typical characters of Calanus euxinus' DVM from a region of a downwelling zone, not reaching the minimum oxygen layer of the Black Sea. Juday net samples are missing to show the DVM of the zooplankton Calanus euxinus and Pseudocalanus elongates, which are claimed to be observed by the acoustics, but one DVM SLs are presents overall in the MS. The swimming speed estimated in the MS was discussed with those of the other studies, but there were no data for the swimming speed of the zooplankton in the results MS. Why other zooplankton which contain the similar body material properties to two targeted copepod species did not appear in the SL of the DVM using the very high frequency, even though the results were discussed for the two mesozooplankton.

Response: Section Discussion is rewritten in line with your and the other reviewers' comments.

Comment #11: Such questions could be clarified already in the MS but the English of the MS is not comprehensible to me.

Response: Prior to submission of the preprint it was edited for proper English language, grammar, punctuation, spelling, and overall style by one or more of the highly qualified native English speaking editors at AJE. This certificate can be verified on the AJE website (aje.com/certificate) using the verification code 1418-EF73-F68E-3B50-E63P.

Please also note the supplement to this comment:
https://os.copernicus.org/preprints/os-2020-106/os-2020-106-AC1-supplement.pdf

---

## Author Comment (AC2) · 10 Mar 2021

Comments to MS os-2020-106. "Annual cycle of sound-scattering mesoplankton in the oxycline and hypoxic zone in the northeastern Black Sea" by Ostrovskii et al. This manuscript presents an interesting dataset taken in the hypoxic zone of the Black Sea. The authors have undoubtedly done a good amount of data collection, analyses, and present their results and conclusions in a reasonable way. However, in its current form, the manuscript still requires further revision according to comments from the another reviewer (RC1) with which I fully agree.

We accomplished a major revision of the manuscript. The most important modifications

are as follows:

1) From the reviewers' comments we learned that many questions can be clarified if the zooplankton sampling data and their analysis are introduced into the manuscript. Hence we invited Dr. Elena Arashkevich who is an expert in the Black Sea mesozoo-plankton to join as a coauthor. In particular, the net sampling data were added into the section Results to validate the sound-scattering layers observed. Other parts of the ms were revised accordingly.

2) Although the concept of mean and median profiles of the sound-scattering data is rather simple we switched to more straightforward approach. We binned all of the acoustic data profiles into the daytime and nighttime groups and computed the daytime and nighttime averaged R-profiles for each observational month to infer the seasonal variability.

3) We additionally analyzed most recent observational data obtained after May 2020.

4) We rewrote both Abstract and Introduction, extended Methods, rearranged and extended Results with 4 new figures, and rewrote both Discussion and Conclusions, References were revised and updated. Certain figures were changed or redrawn. Some material was brought from main body of the manuscript into the Appendix to streamline the research story and to focus on the main points of the ms. Also, mistakes were corrected.

Below, we give our point-by-point answers.

Some aspects that I consider authors should improve, clarify and justify in a better way are: Comment #1: The title is on the annual cycle, but nowhere the annual cycle was actually shown. So, what do authors understand by annual cycle? Authors have taken data from 2013 to 2020 which sounds great, but they have not estimated the annual cycle. I advise authors to perform these calculations in order to affirm the title of their manuscript or simply delete and rewrite it again.

Response: The title of the manuscript was rewritten as follows: Seasonal variation of the sound-scattering zooplankton vertical distribution in the oxygen-deficient waters of the NE Black Sea

Comment #2: The abstract must be completely rewritten since it is not consistent with the classic and logical way of establishing the purpose and motivations of the study, then some background indicating the main results and conclusions.

Respnose: Thank you for pointing this out. The Abstract is rewritten as follows: At the northeastern Black Sea research site, observations of 2010-2020 allowed for study of dynamics and evolution of the mesozooplankton vertical distribution in the oxygen-deficient conditions via analysis of sound-scattering layers associated with dominant zooplankton aggregations. The data were obtained with profiler mooring and zooplankton net sampling. The profiler was equipped with the acoustic Doppler current meter, the conductivity-temperature-depth probe, and fast sensors for the dissolved oxygen [O2]. The acoustic instrument conducted ultrasound (2 MHz) backscatter measurements at 3 angles while being carried by the profiler through the oxic zone. For the lower part of the oxycline and the hypoxic zone, the normalized data of 3 acoustic beams (directional acoustic backscatter ratios, R) indicated the sound-scattering mesozooplankton aggregations, which were described by zooplankton taxonomic and quantitative characteristics based on stratified net sampling at the mooring site. The time series of ~14,000 R-profiles as a function of [O2] at depths where [O2] < 200 $\mu$M were analyzed to determine month-to-month variations of the sound-scattering layers. From spring to early autumn, there were two sound-scattering maxima corresponding to (1) daytime aggregations mainly formed by diel-vertical-migrating copepods Calanus euxinus and Pseudocalanus elongatus and chaetognaths Parasagitta setosa usually at [O2] = 20-90 $\mu$M and (2) persistent monospecific layer of diapausing CV C. euxinus in the suboxic zone at 3 $\mu$M < [O2] < 10 $\mu$M. From late autumn to early winter, no persistent deep sound-scattering layer was observed while the maximum of the daytime mesozooplankton aggregation shifted to the oxygen bounds of 10-30 $\mu$M. At the end

of winter, the acoustic backscatter was basically uniform in the lower part of the oxycline and the hypoxic zone. The assessment of the seasonal variability of the sound-scattering mesozooplankton layers is important for understanding of biogeochemical processes in the oxygen deficient waters.

Comment #3: A weak point in the work is the lack of simultaneous sampling of zooplankton. If this was done, authors should show their time series with these samples to validate the acoustic records. If these samples are already in previous papers then authors should emphasize this point throughout the text. This important aspect is not very clear.

Response: The zooplankton data of several net sampling surveys carried out in different seasons and at different time of the day are used to verify the acoustic records as well as to specify the mesozooplankton species and their biomass in the aggregations in the sea oxygen-deficient zone.

Comment #4: The Introduction section should be rewritten. The backgrounds, motivation, objectives and hypothesis that were tested are missing. For example, paragraph from line 41 should be earlier in the text.

Response: The Introduction is rewritten as you suggested.

Comment #5: Line 70. Correct to "species".

Response : Corrected

Comment #6: Line 92. Please indicate authors or doi (if applicable) rather than insert manual. The same for lines 98 and 99.

Response: The authors are not mentioned in that manual.

Comment #7: Line 120. The authors say ": : : transducer is most sensitive to particles with a diameter of 0.23 mm: : :" then, it is necessary to make a simple comparison between this particle diameter with the respective diameter of the copepod species.

[Figure]

This would help to clarify ideas and further support its acoustic validation.

Response: In the revised ms, the copepod species sizes are defined at the end of the subsection 3.1.

Comment #8: Line 133. Please avoid unconventional symbols or nomenclature. Simply indicate 1 m instead of "10ËĘ0".

Response: Modified throughout the manuscript. Comment #9: Line 143. Do you have observational evidence that the same zooplankton aggregations were sampled? If so, authors should show it as results or if it is an assumption to mention it as such throughout the MS.

Response: Available observational evidence that the same zooplankton aggregations are sampled is included in the revised manuscript.

Comment #10: Line 155. Please indicate accuracy and detection limit for oxygen sensors. This point is important since in several figures there are very low O2 values, so these values are questionable if the detection limit of the instrument is not known. If this is the case, the values below the detection limit must be deleted.

Response: Done. The measurements of the dissolved oxygen using the SBE 43F and Aanderaa 4330F sensors at the moored profiler were carefully studied in the paper by Ostrovskii and Zatsepin: Intense ventilation of the Black Sea pycnocline due to vertical turbulent exchange in the Rim Current area, Deep-Sea Research I, 116, 1–13, doi:10.1016/j.dsr.2016.07.011 (2016) as cited in the revised manuscript. According to the SBE 43F specification the accuracy should be no worse than $\pm2\%$ of saturation as compared with 5% for Aanderaa 4330F that delivers the resolution of < 1 mM or 0.4 %. In practice in the Black Sea, the SBE 43F showed very robust results in detecting the lower boundary of the sea oxic zone consistent with the observations of the sigma-density structure and definition of the oxic zone boundary for the northeastern region of the Sea (Yakushev e al., 2005 among others). In terms of the sensor inertia the SBE

43F outperforms Aanderaa 4330F which is suitable for the purposes of this our study taking into account the profiler vertical speed of nearly 0.2 m/s. It should be noted that every July since 2014, the chemical oceanographers of SIO RAS took samples for determining the oxygen content at the station located to west of our profiler mooring (Dubinin et al., personal communication). We have compared our measurements with the sample data and found that in the suboxic zone two data sets differ by 2-4 mM only with the sampling data showing just slightly higher oxygen while the vertical gradients are essentially the same. We hope to describe the intercomparison data and to present the seasonal change of the oxygen vertical distribution in a separate paper in the future.

Comment #11: Line 179. Please explain in more detail or reference about the hydrogen sulfide zone.

Response: The references are added.

Comment #12: Line 189. The "R ratio" should be explained in more detail. Perhaps one way to start is by inserting an equation and defining the parameters one by one. Furthermore, authors should emphasize the rationale for using this ratio when other methods are available in acoustic measurements.

Response: The equation is inserted and more explanation is added in section Methods.

Comment #13: Line 195. Correct "daythe"

Response: Corrected

Comment #14: Fig. 4. Why do the authors define the hypoxia zone based on density values and not on O2 concentration values? Not bad, but for comparative and conventional purposes it is more useful to define this zone with O2 values.

Response: In the revised section Introduction, the hypoxia zone is defined

Comment #15: Line 206. Delete "compare"

Response: Deleted

Comment #16: Lines 219-227. This paragraph should be in the discussion section.

Response: This paragraph is moved into Discussion.

Comment #17: Figure 7 refers to October 2004... other figures to November 2019. Why this difference? Authors must give justified reasons to present their results in this way.

Response: We rearrange the order of presenting the results.

Comment #18: Figures 8, 9, 11-14. The O2 concentration values are in power format. Please avoid this form and use conventional format. The transformed scale is justified by emphasizing on curves, but numbers not.

Response: The axis labels of these figures are modified.

Comment #19: Figure 9. Is it about the entire time series or some selected months? Please clarify.

Response: The figure caption is modified.

Comment #20: Figure 10. It refers to June-July 2014, why? All these discrepancies in presentation of results into figures without adequate justification and methodological clarification make difficult to understand the main message of the MS.

Response: Thank you for pointing at these discrepancies. We make necessary modifications to address this issue.

Comment #21: Figure 11. Refers to August 2019.

Response: The figure is replaced.

Comment #22: Line 288. Without detailed information on the detection limit of O2 sensors, it is difficult to accept the values indicated here, i.e. 4-9 uM.

Response: The information about the sensor accuracy is added. Please, also see response to the comment10 above.

[Figure]

Comment #23: Lines 318-319. Fluorescence was not measured in this study, if it was, it should be indicated accordingly. If there are support from other sources, they should be indicated.

Response: The work is devoted to the distribution of zooplankton in the lower part of the oxygen zone, so we do not consider the data on chlorophyll.

Comment #24: Figure 14. This figure is fine as a corollary to the main message. I like it, although in 3D view it is somewhat difficult to visualize. Perhaps authors could rotate angles a bit more for better visualization or separate information into two 2D panels.

Response: The figure is modified; however we wonder if it should be included in the revised ms. Probably we would better use it in another paper where we hope to discuss the importance of other environmental factors including the temperature and Chl-a for the dynamics of the mesozooplankton,

Comment #25: Lines 443-444. I do not understand what authors are trying to say in this sentence. Please rewrite.

Response: The sentence is rewritten.

Comment #26: Line 449. Researchgate is a popular and excellent platform to disseminate scientific research; however, to deposit datasets I suggest to use platforms specifically designed for this purpose. Please consider it.

Response: So far we are limited in the options to post the data. However recently, we became involved into the European HORIZON-2020 BRIDGE-BS project and will contribute relevant data into the project data archive in the future.

Comment #27: Finally, consider to reduce the length of the conclusion paragraph.

Response: The section Conclusions is shortened.

Please also note the supplement to this comment:

https://os.copernicus.org/preprints/os-2020-106/os-2020-106-AC2-supplement.pdf

[Figure]

**Supplement:**

Comments to MS os-2020-106. "Annual cycle of sound-scattering mesoplankton in the oxycline and hypoxic zone in the northeastern Black Sea" by Ostrovskii et al. This manuscript presents an interesting dataset taken in the hypoxic zone of the Black Sea. The authors have undoubtedly done a good amount of data collection, analyses, and present their results and conclusions in a reasonable way. However, in its current form, the manuscript still requires further revision according to comments from the another reviewer (RC1) with which I fully agree.

We accomplished a major revision of the manuscript. The most important modifications are as follows:

1) From the reviewers' comments we learned that many questions can be clarified if the zooplankton sampling data and their analysis are introduced into the manuscript. Hence we invited Dr. Elena Arashkevich who is an expert in the Black Sea mesozooplankton to join as a coauthor. In particular, the net sampling data were added into the section Results to validate the sound-scattering layers observed. Other parts of the ms were revised accordingly.

2) Although the concept of mean and median profiles of the sound-scattering data is rather simple we switched to more straightforward approach. We binned all of the acoustic data profiles into the daytime and nighttime groups and computed the daytime and nighttime averaged R-profiles for each observational month to infer the seasonal variability.

3) We additionally analyzed most recent observational data obtained after May 2020.

4) We rewrote both Abstract and Introduction, extended Methods, rearranged and extended Results with 4 new figures, and rewrote both Discussion and Conclusions, References were revised and updated. Certain figures were changed or redrawn. Some material was brought from main body of the manuscript into the Appendix to streamline the research story and to focus on the main points of the ms. Also, mistakes were corrected.

Below, we give in blue our point-by-point answers.

Some aspects that I consider authors should improve, clarify and justify in a better way are:

1) The title is on the annual cycle, but nowhere the annual cycle was actually shown. So, what do authors understand by annual cycle? Authors have taken data from 2013 to 2020 which sounds great, but they have not estimated the annual cycle. I advise authors to perform these calculations in order to affirm the title of their manuscript or simply delete and rewrite it again.

Response: The title of the manuscript was rewritten as follows:

Seasonal variation of the sound-scattering zooplankton vertical distribution in the oxygen-deficient waters of the NE Black Sea

2)  The abstract must be completely rewritten since it is not consistent with the classic and logical way of establishing the purpose and motivations of the study, then some background indicating the main results and conclusions.

Thank you for pointing this out. The Abstract is rewritten as follows:

At the northeastern Black Sea research site, observations of 2010-2020 allowed for study of dynamics and evolution of the mesozooplankton vertical distribution in the oxygen-deficient conditions via analysis of sound-scattering layers associated with dominant zooplankton aggregations. The data were obtained with profiler mooring and zooplankton net sampling. The profiler was equipped with the acoustic Doppler current meter, the conductivity-temperature-depth probe, and fast sensors for the dissolved oxygen [O2]. The acoustic instrument conducted ultrasound (2 MHz) backscatter measurements at 3 angles while being carried by the profiler through the oxic zone. For the lower part of the oxycline and the hypoxic zone, the normalized data of 3 acoustic beams (directional acoustic backscatter ratios, R) indicated the sound-scattering mesozooplankton aggregations, which were described by zooplankton taxonomic and quantitative characteristics based on stratified net sampling at the mooring site. The time series of ~14,000 R-profiles as a function of [O2] at depths where [O2] < 200 μM were analyzed to determine month-to-month variations of the sound-scattering layers. From spring to early autumn, there were two sound-scattering maxima corresponding to (1) daytime aggregations mainly formed by diel-vertical-migrating copepods Calanus euxinus and Pseudocalanus elongatus and chaetognaths Parasagitta setosa usually at [O2] = 20-90 μM and (2) persistent monospecific layer of diapausing CV C. euxinus in the suboxic zone at 3 μM < [O2] < 10 μM. From late autumn to early winter, no persistent deep sound-scattering layer was observed while the maximum of the daytime mesozooplankton aggregation shifted to the oxygen bounds of 10-30 μM. At the end of winter, the acoustic backscatter was basically uniform in the lower part of the oxycline and the hypoxic zone. The assessment of the seasonal variability of the sound-scattering mesozooplankton layers is important for understanding of biogeochemical processes in the oxygen deficient waters.

3)  A weak point in the work is the lack of simultaneous sampling of zooplankton. If this was done, authors should show their time series with these samples to validate the acoustic records. If these samples are already in previous papers then authors should emphasize this point throughout the text. This important aspect is not very clear.

The zooplankton data of several net sampling surveys carried out in different seasons and at different time of the day are used to verify the acoustic records as well as to specify the mesozooplankton species and their biomass in the aggregations in the sea oxygen-deficient zone.

4) The Introduction section should be rewritten. The backgrounds, motivation, objectives and hypothesis that were tested are missing. For example, paragraph from line 41 should be earlier in the text.

The Introduction is rewritten as you suggested.

5) Line 70. Correct to "species".

Corrected

6) Line 92. Please indicate authors or doi (if applicable) rather than insert manual. The same for lines 98 and 99.

The authors are not mentioned in that manual.

7) Line 120. The authors say ": : : transducer is most sensitive to particles with a diameter of 0.23 mm: : :" then, it is necessary to make a simple comparison between this particle diameter with the respective diameter of the copepod species. This would help to clarify ideas and further support its acoustic validation.

In the revised ms, the copepod species sizes are defined at the end of the subsection 3.1.

8) Line 133. Please avoid unconventional symbols or nomenclature. Simply indicate 1 m instead of "10ˆ0".

Modified throughout the manuscript.

9) Line 143. Do you have observational evidence that the same zooplankton aggregations were sampled? If so, authors should show it as results or if it is an assumption to mention it as such throughout the MS.

Available observational evidence that the same zooplankton aggregations are sampled is included in the revised manuscript.

10) Line 155. Please indicate accuracy and detection limit for oxygen sensors. This point is important since in several figures there are very low O2 values, so these values are questionable if the detection limit of the instrument is not known. If this is the case, the values below the detection limit must be deleted.

Done. The measurements of the dissolved oxygen using the SBE 43F and Aanderaa 4330F sensors at the moored profiler were carefully studied in the paper by Ostrovskii and Zatsepin: Intense ventilation of the Black Sea pycnocline due to vertical turbulent exchange in the Rim Current area, Deep-Sea Research I, 116, 1–13, doi:10.1016/j.dsr.2016.07.011 (2016) as cited in the revised manuscript. According to

the SBE 43F specification the accuracy should be no worse than ±2% of saturation as compared with 5% for Aanderaa 4330F that delivers the resolution of < 1 mM or 0.4 %. In practice in the Black Sea, the SBE 43F showed very robust results in detecting the lower boundary of the sea oxic zone consistent with the observations of the sigma-density structure and definition of the oxic zone boundary for the northeastern region of the Sea (Yakushev e al., 2005 among others). In terms of the sensor inertia the SBE 43F outperforms Aanderaa 4330F which is suitable for the purposes of this our study taking into account the profiler vertical speed of nearly 0.2 m/s. It should be noted that every July since 2014, the chemical oceanographers of SIO RAS took samples for determining the oxygen content at the station located to west of our profiler mooring (Dubinin et al., personal communication). We have compared our measurements with the sample data and found that in the suboxic zone two data sets differ by 2-4 mM only with the sampling data showing just slightly higher oxygen while the vertical gradients are essentially the same. We hope to describe the intercomparison data and to present the seasonal change of the oxygen vertical distribution in a separate paper in the future.

11) Line 179. Please explain in more detail or reference about the hydrogen sulfide zone.
   The references are added.

12) Line 189. The "R ratio" should be explained in more detail. Perhaps one way to start is by inserting an equation and defining the parameters one by one. Furthermore, authors should emphasize the rationale for using this ratio when other methods are available in acoustic measurements.
   The equation is inserted and more explanation is added in section Methods.

13) Line 195. Correct "daythe"
   Corrected

14) Fig. 4. Why do the authors define the hypoxia zone based on density values and not on O2 concentration values? Not bad, but for comparative and conventional purposes it is more useful to define this zone with O2 values.
   In the revised section Introduction, the hypoxia zone is defined

15) Line 206. Delete "compare"
   Deleted

16) Lines 219-227. This paragraph should be in the discussion section.
   This paragraph is moved into Discussion.

17) Figure 7 refers to October 2004... other figures to November 2019. Why this difference? Authors must give justified reasons to present their results in this way.

We rearrange the order of presenting the results.

18) Figures 8, 9, 11-14. The O2 concentration values are in power format. Please avoid this form and use conventional format. The transformed scale is justified by emphasizing on curves, but numbers not.

The axis labels of these figures are modified.

19) Figure 9. Is it about the entire time series or some selected months? Please clarify.

The figure caption is modified.

20) Figure 10. It refers to June-July 2014, why? All these discrepancies in presentation of results into figures without adequate justification and methodological clarification make difficult to understand the main message of the MS.

Thank you for pointing at these discrepancies. We make necessary modifications to address this issue.

21) Figure 11. Refers to August 2019.

The figure is replaced.

22) Line 288. Without detailed information on the detection limit of O2 sensors, it is difficult to accept the values indicated here, i.e. 4-9 uM.

The information about the sensor accuracy is added. Please, also see response to the comment10 above.

23) Lines 318-319. Fluorescence was not measured in this study, if it was, it should be indicated accordingly. If there are support from other sources, they should be indicated.

The work is devoted to the distribution of zooplankton in the lower part of the oxygen zone, so we do not consider the data on chlorophyll.

24) Figure 14. This figure is fine as a corollary to the main message. I like it, although in 3D view it is somewhat difficult to visualize. Perhaps authors could rotate angles a bit more for better visualization or separate information into two 2D panels.

The figure is modified; however we wonder if it should be included in the revised ms. Probably we would better use it in another paper where we hope to discuss the importance of other environmental factors including the temperature and Chl-a for the dynamics of the mesozooplankton,

25) Lines 443-444. I do not understand what authors are trying to say in this sentence. Please rewrite.

The sentence is rewritten.

25) Line 449. Researchgate is a popular and excellent platform to disseminate scientific research; however, to deposit datasets I suggest to use platforms specifically designed for this purpose. Please consider it.

So far we are limited in the options to post the data. However recently, we became involved into the European HORIZON-2020 BRIDGE-BS project and will contribute relevant data into the project data archive in the future.

26) Finally, consider to reduce the length of the conclusion paragraph.

The section Conclusions is shortened.

---

## Author Comment (AC3) · 10 Mar 2021

Ostrovskii et al. Anonymous Referee #3 The manuscript (MS) presents a modern imaging techniques such as the acoustic of pelagic communities with advantages to be informative about heterogeneity and transcend multiple spatial scales. The article is based on a large data set (2013-2020) obtained from the application of an alternative innovative approach - a moored Aqualog profiler equipped with an ultrasound probe, a conductivity-temperature-depth (CTD) probe, and a fast

oxygen sensor with the advantage of frequent year-round measurements of collocated vertical profiles of sound scattering, temperature, salinity, and oxygen concentration in the water column from the near-surface to the bottom layer with a high vertical resolution. This topic is not novel but the previous studies are based on ship-borne echograms. The authors clearly indicate their own original contribution. Printer-friendly version Discussion paper The work is interesting, results are sufficient and the paper addresses scientific questions within the scope of OS but needs some revisions.

Response: We are grateful to reviewer for the comments. In the following, we give our point-by-point answers.

Comment #1: The abstract should be condensed and concentrated around the main aim, results and conclusions.

Response: The abstract is condensed although the new information was added to reflect new important contribution about the acoustic data verification based on the zooplankton net sampling.

Comment #2: In the introduction the main sound-scattering zones are defined according to Ostrovskii and Zatsepin (2011) but I suggest to bind them with the density sigma theta which is relevant to the mesozooplankton vertical distribution especially for the Black Sea. As a consequence, it needs to be developed and compared in the results and discussion chapters.

Response: This was done. In the section Results, more information about the isopycnal surfaces is added into the figures, also the new Fig. 11 is added to compare the depth profile of R with the sigma profile of R.

Comment #3: In the MS the lowest depth mentioned was at _Ït' = 15.9 kg m-3. However, in other studies (Mutlu 2007a, b,) sigma theta - 16.2 kg.m-3 , identified as oxygen minimum zone (OMZ) (Tugrul et al. 1992), is a layer where Calanus euxinus spend their daytime. How will the authors comment these differences?

Response: There are regional differences in the lower boundary of the oxygen zone in the Black Sea as it was shown by Glazer et al. (2006a, 2006b). In the southern regions of the Sea adjacent to the Bosphorus strait, the Sea is ventilated due to the inflow of the Mediterranean water. According to Galzer et al. (2006a), "Layers of oxygen intrusion (5 m thick, from 10 to 150 mM O₂) were present within the suboxic zone of the southwest Black Sea that are not present in the west-central and northeast Black Sea. Oxygen injection also occurs at other depths throughout the southwest and corresponds with small temperature anomalies, suggesting influence by Bosphorus inflow up to 150 km from its entrance to the Black Sea." Also according to Glazer et al. (2006b) there are year-to-year-variations in the southwest region as follows: "We observed much less lateral oxygen injection from the Bosphorus in 2003 (less than 95 km from Bosphorus) than in 2001 (up to 150 km). This difference can be attributed to variability in physical processes including seasonal temperature and wind variations between winter conditions (2003) and early summer conditions (2001). Furthermore, suboxic zone thickness varied basin-wide, exhibiting changes in the depth of oxygen extinction and sulfide onset." As concerns with the northeastern Black Sea, the oxygen disappearance was reported for the isopycnal 15.9 (Ostrovskii and Zatsepin, 2016).

Comment #4: The authors presented different seasonal variation in mesoplankton dynamics in relation to dissolved oxygen concentrations. Additionally the SL amplitude showed differences in same months but a reasonable explanation is not presented.

Response: It seems that the difference you noted for the same months is due to the year-to-year variations in the mesozooplankton abundance.

Comment #5: There are two dominant species well acoustically discriminated in the Black Sea – Calanus euxinus and Parasagitta setosa (Mutlu 2007) but the later was not included in the MS which need an explanation.

Response: This is addressed by adding available data of zooplankton sampling nearby

the profiler mooring. The figures 4, 6, 8, and 10 in the revised manuscript show the biomass data for Parasagitta setosa.

Comment #6: Line 315 The authors say ": : :two layers in the cold intermediate layer (CIL) (temperature less than 9_C),::" but according to the literature the positions of the 8_C isotherms have traditionally been considered the lower and upper boundaries of the CIL (Blatov et al., 1984; Ozsoy and Unluata, 1997). Winter cooling, which is an essential element of the seasonal variability could be used for comparison of unlike SL profiles in the same season (month) in different years.

Response: The cold intermediate layer was getting significantly warmer recently. According to (Stanev, E. V., Peneva, E., & Chtirkova, B. (2019). Climate change and regional ocean water mass disappearance: Case of the Black Sea. Journal of Geophysical Research: Oceans, 124, 4803–4819. https://doi.org/10.1029/2019JC015076) "Data from profiling [ARGO] floats reveal that climate change in the Black Sea leads to the disappearance of specific water masses. The warming trend in the cold intermediate layer (CIL) of ∼0.05 °C/year was more than double the trend in previous decades, and its temperature approached that of the waters in the deeper layers (∼9 °C), which signified its disappearance. This evolution was due to the warmer winters over the last 14 years. Intermittent major cold water formation events (only three during this period) could not sufficiently refill the CIL."

Comment #7: Conclusions should be rewritten - shortened, concentrated and clearer, emphasizing the research contribution.

Response: The section Conclusions is rewritten in line with your comment.

Comment #8: Correction: Pseudocalanus elongatus (WoRMS) is the right species name, not Pseudocalanus elongates.

Response: Sorry for this mistake. It is corrected.

Comment #9: Figure 3 It is mentioned that "The horizontal axis represents UTC time."

Please, check.

Response: This is corrected.

Comment #10: References should be checked. For example, Arashkevich et al. 2014 (in the text) Arashkevich et al. 2013 (in the reference list); Arashkevich et al. 199, Besiktepe et al., 1998 are missing in the reference list but are cited in the MS and etc.

Response: The missing references are added.

Comment #11: The language should be precise.

Response: We tried to do our best when revised the ms. We also noticed that the journal processing charges include English language copy-editing for final revised papers. We hope that if the ms is accepted it will be edited for precise English language.

Please also note the supplement to this comment:
https://os.copernicus.org/preprints/os-2020-106/os-2020-106-AC3-supplement.pdf

––––––––––––––––––––––––––––

---

## Referee Report (RR1)

**Comments to MS os-2020-106. "Annual cycle of sound-scattering mesoplankton in the oxycline and hypoxic zone in the northeastern Black Sea" by Ostrovskii et al.**

In this manuscript authors present an analysis of the seasonal variability of the sound-scattering mesozooplankton layers, on the basis of acoustic data (2 MHz ADCP), oxygen and other physical parameters from a moored automatic mobile profiler. The influence of zooplankton diel vertical migration limited by the oxygen concentration, and vertical distributions of temperature and density are considered. Net sampling data confirmed existence of migrating and non-migrating zooplankton at the study location. Although the topic has been addressed earlier by several authors, it presents a new data set, analyses, results and conclusions. Also, previous studies are adequately mentioned and referenced in the manuscript, and the new outcomes are well highlighted.

**Comments**

1) Figure 9. It is mentioned that "The slope of a straight arrow pointing downwards corresponds to a diving speed of ~ 1.5 cm s-1. The ascent is accelerated and reaches values of approximately 2.5 cm s-1 in the upper 60 m depth." However, the method for calculating vertical (ascent and descent) swimming speeds is not described in the manuscript, so it is hard to test the validity of these results. The authors should explain how migration speeds are calculated.

2) Figures 4, 6, 8 and 10. There is a discrepancy between the legend and the text to the figures: "CE – Calanus euxinus; PE – Pseudocalanus elongatus; SC – small crustaceans; NS – Noctiluca scintillans; PE – Parasagitta setosa; Var – varia". Please check.

3) "Appendix A" should be summarised and referred in the body of the manuscript.

4) Lines 97, 98, and 99. Please indicate the company name and the year of publication rather than insert a hyperlink to the manual.

---

## Author Response (AR2)

Dear Prof. Hoppema:

Thank you very much for your careful editing of our manuscript. We feel that the paper became much better than the original preprint.

Please, find below a point-by-point reply to your comments. Notice that our revised manuscript contains the figures where we also made changes such as the usage of acronym LT for the local time, and the date format as you suggested.

My best regards on behalf of our team of the paper authors,

Alexander Ostrovskii

Point-by-point response to the comments:

L10 [O2] means the concentration of oxygen. So here it should be either: for the concentration of dissolved oxygen [O2] or: … for dissolved oxygen (O2).

Author's response - "The concentration of dissolved oxygen" was used.

L18 „diapausing CV C. euxinus" This is not clear. CV C.? This may be sound knowledge for planktologists, but in the abstract please explain.

Author's response - Replaced by "the diapausing fifth copepodite stages of Calanus euxinus"

L25 Is polluted the right word here? It is not caused by anthropogenic activity, is it?

Author's response - Replaced by "contaminated".

L30 Please define CTD here as it is used for the first time

Author's response - The CTD was defined.

L31-32 Vinogradov and Nalbandov, 1990; L49 Erkan and Gücü ; please add "and" between the names in all cases in the manuscript where the citation concerns two authors.

Author's response - Modified in all cases in the manuscript.

L223 21-22 June 2010 (format) Please change to this format through the manuscript at many places

Author's response - Modified in all cases in the manuscript.

L232 58±14% Is this standard deviation or standard error? Please explain for all results.

Author's response - The explanation was given.

Fig 11 (and other places) Please use LT for local time.

Author's response - Modified in all cases in the manuscript.

L425 species not spices

Author's response - Corrected.

L579 Please correct: 3÷10 µM What is meant here?

Author's response - Corrected.

---

## Author Response (AR3)

Dear Prof. Hoppema:

Thank you for allowing us the opportunity to respond to the peer reviews. The reviewers' suggestions were addressed in the section Measurements as follows (in blue ink).

Suggestion #1 for revision by Anonymous Referee #1 - The method for calculating vertical (ascent and descent) swimming speeds of zooplankton should be describe in the text (Measurements section).

Response by the authors  - The paragraph was added at LL. 232-236:  "One method for calculating vertical migration speed of zooplankton from the sound backscatter data of the acoustic current meter at the profiler Aqualog was described in (Pezacki et al., 2017). However the vertical migration speed of mesozooplankton is beyond the focus of this study. Only once when discussing the pattern of the diel vertical migration, the slope of the migration track on the echogram (see Figure 9 below) is considered to give rough idea about the dive and the ascend of  mesozooplnakton. Much more effort would certainly be needed to visualize the specimens' vertical swimming."

Suggestion #2 for revision by Anonymous Referee #1 - "Appendix A" should be summarised and referred in the manuscript.

Response by the authors – The following changes were made LL. 232-236:  "In Appendix, we will consider whether the mesozooplankton specimens' vertical orientation is tilted in the deep aggregations. The analysis will be based on calculation of the ratio of the volume scattering strength of the horizontal beams $A_1/A_2$  assuming that due to the tilt the standard deviation of $A_1/A_2$   should be greater than 0."

Suggestion #1 for revision by Anonymous Referee #2 -  My only observation to this version is that I still do not see an adequate presentation and justification for the dates shown within the time series that authors have collected. For example, authors indicate that they sampled the hydrographic features 15 times but show sections from 2016 onwards. On the other hand, zooplankton sampling was carried out on 4 dates/years. Why are these discrepancies observed? Any potential limitations that authors can discuss in this regard? It's probably not a big problem but at least for me it's not entirely clear. Perhaps a table could help clarify this minor point so that it is easily interpreted.

Response by the authors  –  Yes, it is really worth to add the time table of the observational surveys. The table entitled Deployments of the Profiler Aqualog-6 with Nortek Aquadopp Current Meter in the NE Black Sea and the Dates of the Zooplankton Sampling near the Profiler Mooring Site in 2010-2021 was added (please, see LL. 202-206 and below in the revised manuscript).

Finally, please, also notice that we updated Figure 15 of the revised manuscript.

Best regards,
Alexander Ostrovskii